# Linker histone H1.2 and H1.4 affect the neutrophil lineage determination

**Gabriel Sollberger[1,2]\*, Robert Streeck[1,3], Falko Apel[1,3], Brian Edward Caffrey[4†‡], Arthur I Skoultchi[5], Arturo Zychlinsky[1]**

[1]Max Planck Institute for Infection Biology, Department of Cellular Microbiology, Berlin, Germany; [2]University of Dundee, School of Life Sciences, Division of Cell Signalling and Immunology, Dundee, United Kingdom; [3]Institut für Biologie, Humboldt Universität zu Berlin, Berlin, Germany; [4]Max Planck Institute for Molecular Genetics, Berlin, Germany; [5]Department of Cell Biology, Albert Einstein College of Medicine, Bronx, United States

**\*For correspondence:**
gsollberger001@dundee.ac.uk

**Present address:** [†]Berlin Institute of Health (BIH), Berlin, Germany; [‡]Charité-Universitätsmedizin Berlin, corporate member of Freie Universität Berlin, Humboldt-Universität zu Berlin, Berlin Institute of Health, Berlin, Germany

**Competing interests:** The authors declare that no competing interests exist.

**Abstract** Neutrophils are important innate immune cells that tackle invading pathogens with different effector mechanisms. They acquire this antimicrobial potential during their maturation in the bone marrow, where they differentiate from hematopoietic stem cells in a process called granulopoiesis. Mature neutrophils are terminally differentiated and short-lived with a high turnover rate. Here, we show a critical role for linker histone H1 on the differentiation and function of neutrophils using a genome-wide CRISPR/Cas9 screen in the human cell line PLB-985. We systematically disrupted expression of somatic H1 subtypes to show that individual H1 subtypes affect PLB-985 maturation in opposite ways. Loss of H1.2 and H1.4 induced an eosinophil-like transcriptional program, thereby negatively regulating the differentiation into the neutrophil lineage. Importantly, H1 subtypes also affect neutrophil differentiation and the eosinophil-directed bias of murine bone marrow stem cells, demonstrating an unexpected subtype-specific role for H1 in granulopoiesis.

## Introduction

Because multicellular organisms often face disturbances in their homeostasis, they evolved complex immune responses to deal with sterile and infectious insults. Like elsewhere in the human body, immune cells differentiate from multipotent precursors. In the immune context, precursors are primarily hematopoietic stem cells in the bone marrow. One of these differentiation programs, called granulopoiesis, leads to the development of granulocytes - basophils, eosinophils and neutrophils. Neutrophils are the most abundant leukocytes in humans. In homeostatic conditions, up to $2 \times 10^{11}$ neutrophils enter the blood stream per day and patrol the host's body until they sense signs of infection, which triggers them to leave the blood stream and migrate to the inflammatory site where they ensure pathogen removal. An efficient neutrophil response is crucial for human antimicrobial defense and, correspondingly, neutropenia is associated with severe infections (**Klein, 2011**).

The three best described defense mechanisms of neutrophils are phagocytosis (engulfment of pathogenic microorganisms and subsequent destruction in phagosomes), degranulation (the release of antimicrobial proteins from granules to the extracellular space) and the formation of neutrophil extracellular traps (NETs). NETs are chromatin structures studded with antimicrobial proteins derived mostly from neutrophil granules, which can trap pathogens (**Papayannopoulos, 2018**). Neutrophils release NETs via various mechanisms, which in most cases lead to neutrophil death (**Kenny et al., 2017**); thus, NETs are an extreme form of antimicrobial defense, as their release also eliminates the cell casting the NET. Under homeostatic conditions, NET formation occurs in mature, circulating neutrophils and therefore we propose that specific cues during neutrophil differentiation shape their

ability to form NETs. It is noteworthy that during systemic inflammation, some neutrophils, called low density granulocytes, display signs of immaturity, but enhanced formation of NETs (*Denny et al., 2010*; *Villanueva et al., 2011*). In inflammatory settings the requirement for full differentiation might therefore be less stringent.

Neutrophils differentiate from granulocyte-macrophage precursors. The first progenitor with a clear neutrophil lineage preference during granulopoiesis is the promyelocyte (*Yvan-Charvet and Ng, 2019*). During maturation, neutrophils acquire a lobulated nuclear shape and produce many of their antimicrobial effector proteins, stored in granules (*Borregaard, 2010*). Granulopoiesis requires balanced activity of various transcription factors (*Yvan-Charvet and Ng, 2019*). The C/EBP family, especially C/EBP-α and –ε, are key regulators of neutrophil differentiation (*Zhang et al., 1997*; *Yamanaka et al., 1997*). Other transcription factors can antagonize C/EBP-dependent neutrophil development. For example, expression of GATA-1 (*Yu et al., 2002*) and GATA-2 (*Hirasawa et al., 2002*) induce eosinophil, basophil or mast cell differentiation (*Fiedler and Brunner, 2012*).

Differentiation programs rely both on transcription factor activation and on epigenetic chromatin alterations. DNA wraps around an octamer of core histones forming a nucleosome. The linker histone H1 binds to nucleosomes and further compacts chromatin (*Bednar et al., 2017*). There are 11 different H1 subtypes in mice and humans. Five of them (H1.1 – H1.5) are somatic replication-dependent subtypes, two (H1.0 and H1X) are somatic and replication-independent and four (H1oo, H1t, H1T2 and HILS1) are germline-specific (*Hergeth and Schneider, 2015*). All H1 subtypes consist of a highly conserved globular domain and two more variable regions: the N- and the C-terminal tails. Individual H1 subtypes are more conserved between species than H1 subtypes are within one species, suggesting that the conservation of different subtypes has functional relevance (*Hergeth and Schneider, 2015*). Still, some functions of H1, such as global shaping of chromatin structure, are redundant between subtypes. This is reflected by the fact that mice deficient for one or even two H1 subtypes are viable and fertile without striking morphological abnormalities (*Fan et al., 2003*). However, mice lacking three H1 subtypes die in utero, with total H1 levels reduced to about 50% (*Fan et al., 2003*). Appropriate H1 expression is therefore essential for development. Despite some redundancy, H1 subtypes have specific functions. H1 subtypes bind to chromatin with different affinities and knock down of specific H1 subtypes affects gene transcription in distinct ways (*Hendzel et al., 2004*; *Th'ng et al., 2005*; *Sancho et al., 2008*).

The role of H1 and H1 subtypes in immunity is poorly understood. H1 contributes to the silencing of pro-inflammatory cytokines after endotoxin challenge (*El Gazzar et al., 2009*), as well as to the silencing of interferon-stimulated genes (*Kadota and Nagata, 2014*). Interestingly, mice deficient in H1.0 have fewer dendritic cells, but normal numbers of granulocytes, macrophages and lymphocytes, showing a subtype-specific role for H1 in dendritic cell differentiation (*Gabrilovich et al., 2002*). However, to date, no studies have systematically addressed H1 subtype involvement in the maturation and function of human immune cells.

Here we used the human neutrophil-like cell line, PLB-985, to study neutrophil maturation and function. We performed a genome-wide CRISPR/Cas9 screen using survival to a NET-inducing stimulus, phorbol 12-myristate 13-acetate (PMA), as a readout. Surprisingly, we found that depletion of either H1.2 or H1.4 strongly reduced the differentiation of PLB-985 and thereby their ability to form NETs. Notably, deficiency of the other somatic H1 subtypes, H1.1, H1.3 and H1.5, accelerated PLB-985 maturation. RNA-seq analysis showed that H1.2 and H1.4 deficiency leads to an upregulation of eosinophil genes in PLB-985. Mirroring these results, murine bone marrow stem cells from H1.2/H1.4 double deficient mice shifted their differentiation profile from neutrophil towards eosinophil cell fate, whereas neutrophils in vivo displayed disturbances in ageing markers. We further demonstrate that the subtype-specific function of H1 in neutrophil and eosinophil differentiation depends – at least in part – on the transcription factor GATA-2. We uncovered that, unexpectedly, H1 subtypes affect lineage specification during granulopoiesis.

## Results

### PLB-985: a model for neutrophil maturation and function

We used the human promyelocytic cell line PLB-985 (*Tucker et al., 1987*). PLB-985 differentiate into neutrophil-like cells that resemble primary human neutrophils in various ways (*Zhen et al., 1993*;

*Pedruzzi et al., 2002*; *Pivot-Pajot et al., 2010*; *Marin-Esteban et al., 2012*). PLB-985 changed their nuclear morphology and developed granules, though fewer than primary neutrophils, throughout the 7 days of differentiation (referred as d0 to d7 in this report) (*Figure 1a*). Once differentiated, PLB-985 also produced ROS in response to the agonist phorbol 12-myristate 13-acetate (PMA) (*Figure 1b*, *Figure 1—figure supplement 1a*) and upregulated the surface marker CD11b (*Figure 1c*). Furthermore, differentiated PLB-985 phagocytosed *Escherichia coli*, a process that could be blocked by the phagocytosis inhibitor cytochalasin B (*Figure 1—figure supplement 1b*). Importantly, PMA potently induced cell death in fully differentiated PLB-985, but not in d0 or d3 cells (*Figure 1d*). PMA-induced cell death was morphologically similar to NETosis; dying PLB-985 expanded their nuclei and a significant fraction released DNA as shown by SYTOX Green staining (*Figure 1e*, *Figure 1—figure supplement 1c*). Of note, the calcium ionophore A23187, another stimulant of NET formation, also induced cell death in PLB-985, both at d3 and d7 of differentiation (*Figure 1—figure supplement 1d*). Taken together, these experiments show that PLB-985 cells differentiate into neutrophil-like phagocytes that produce ROS and NETs.

To show that PLB-985 can be genetically modified by CRISPR/Cas9, we disrupted expression of NOX2 (*CYBB*), a component of the NADPH oxidase complex, which is required for PMA-induced NETosis (*Fuchs et al., 2007*). We also transduced PLB-985 with a non-target scrambled single-guide RNA (sgRNA) as a control (subsequently called scr.). Differentiated PLB-985 cells lacking NOX2 failed to produce an oxidative burst and did not undergo cell death in response to PMA (*Figure 1—figure supplement 1e,f*). To inactivate NADPH oxidase independently of NOX2, we also disrupted expression of its NCF2 subunit, and showed again that these cells did not die upon PMA stimulation (*Figure 1—figure supplement 1g*). These results demonstrate that PLB-985, which can be genetically modified, are a suitable model to mimic human neutrophils.

## A genome-wide screen to identify regulators of neutrophil maturation and function

We found that PMA-induced NET formation required full differentiation of PLB-985. Therefore, we used this readout to perform a genome-wide CRISPR/Cas9 screen to identify genes required for neutrophil maturation and function (*Figure 1f*). We reasoned that, besides NET-defective cells, we could also identify differentiation-defective cells by sorting cells that survived PMA treatment. We transduced PLB-985 with a library targeting all human genes with 5–6 sgRNAs per gene (*Shalem et al., 2014*). As expected, PMA stimulation reduced the fraction of identified guides in the survivor population (*Figure 1—figure supplement 1h*), indicating selection of specific clones.

We defined 'hits' as genes for which at least 50% of the identified sgRNAs were overrepresented (more than two-fold) in our survivor population. Underlining the validity of the screening approach, we identified *CYBB*, *NCF1* and *MPO* as hits; all three are known to be required for PMA-induced NET formation (*Fuchs et al., 2007*; *Metzler et al., 2011*; *Figure 1g*, *Figure 1—source data 1*). We also found *CEBPD*, a member of the C/EBP family of transcription factors that are crucial for granulopoiesis (*Wang and Friedman, 2002*; *Figure 1g*). Unexpectedly, we identified two members of the linker histone H1 family among the 20 highest ranked genes. *HIST1H1E* (H1.4) was overrepresented with 100% of sgRNAs and showed the highest median enrichment (*Figure 1g*, *Figure 1—source data 1*). Notably, there are no studies showing a functional role of H1 in neutrophil differentiation or NET formation.

## H1.4 and H1.2 affect neutrophil function through differentiation

There are five somatic replication-dependent (H1.1 – H1.5) and two replication-independent H1 subtypes (H1.0 and H1X). PLB-985 expressed mRNA encoding H1.2 and H1.4 more abundantly than the other subtypes (*Figure 2a*). The expression of replication-dependent H1 subtypes occurs mainly in the S phase of proliferating cells and, accordingly, H1 mRNA levels decreased during PLB-985 differentiation (*Figure 2—figure supplement 1a*, same experiments as in *Figure 2a*, but plotted as relative to d0). As expected, this reduction was more evident for subtypes with a higher baseline expression (*Figure 2a*, *Figure 2—figure supplement 1a*). When we looked at H1.2 and H1.4 protein expression of PLB-985 during differentiation, we found that protein levels also decreased as the cells matured (*Figure 2—figure supplement 1b*).

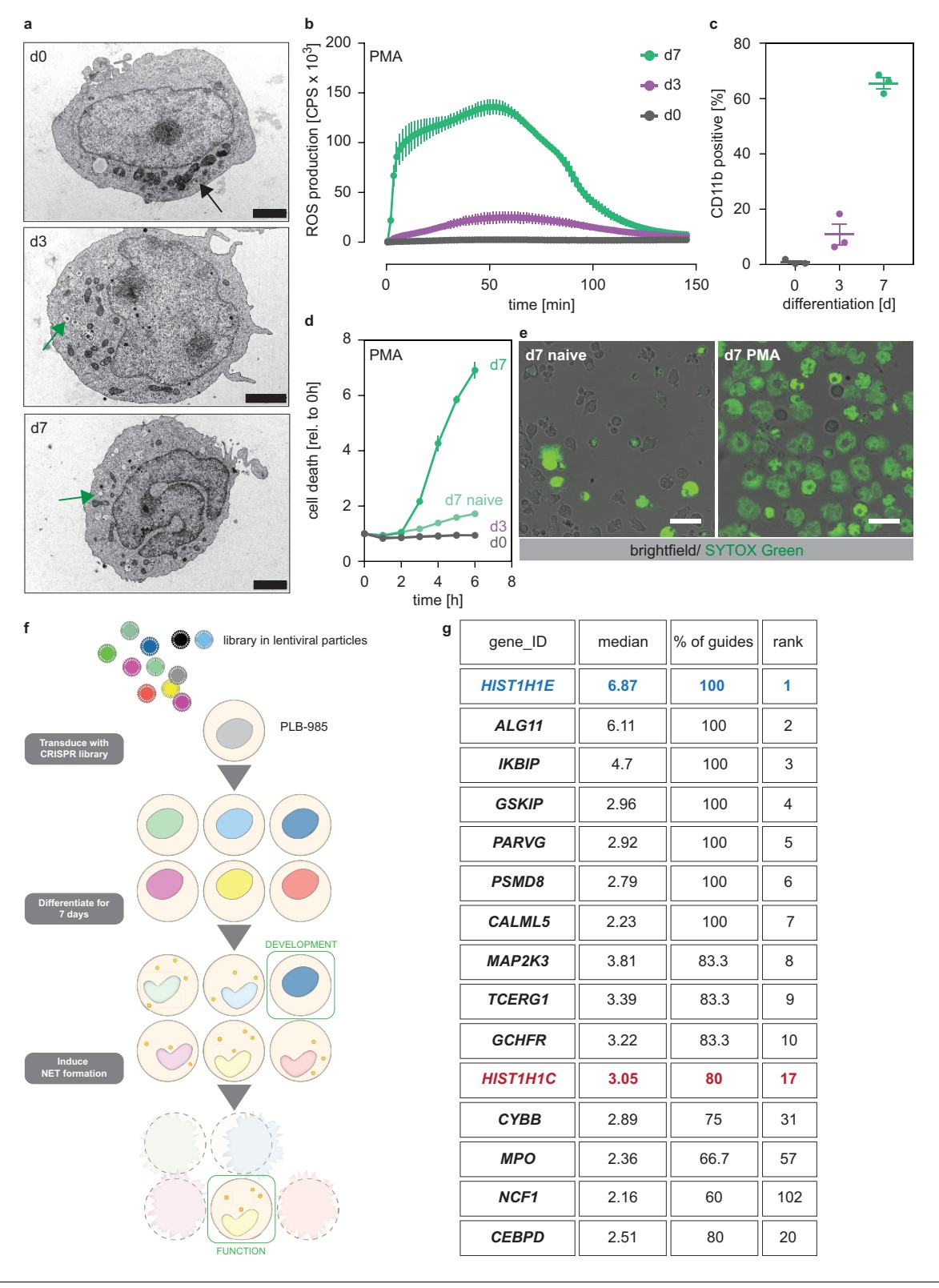

**Figure 1.** A genome-wide screen to identify genes required for PLB-985 cells differentiation and function. (a–d) Characterization of PLB-985 differentiation and function at d0, d3 and 7 of differentiation. (a) Electron microscopy images of PLB-985 showing acquisition of granules (green arrows, the black arrow at d0 indicates mitochondria) and changes in nuclear morphology. Scale bars correspond to 5 µm. (b) ROS production of PLB-985 in response to 100 nM PMA, showing that only fully differentiated PLB-985 produced an oxidative burst. (c) Surface expression of CD11b, depicted is the

*Figure 1 continued on next page*

*Figure 1 continued*

percentage of CD11b positive cells out of all viable singlets. (**d**) PMA-induced cell death was measured over time after addition of the cell-impermeable DNA dye SYTOX Green and analysis of fluorescence indicating cell death. 'd7 naïve' indicates differentiated cells, which were not treated with PMA. (**e**) Representative images of differentiated PLB-985 (d7) after addition of SYTOX Green and treatment with or without PMA. Scale bars are 20 μm. (**f**) Outline of the CRISPR/Cas9 screen. Cells were transduced with a genome-wide CRISPR/Cas9 library (lentiviral particles, small circles), differentiated for 7 days, treated with PMA for 16 hr and survivors were sorted and sequenced to identify sgRNAs. (**g**) Top 10 of the screen, H1.2 (*HIST1H1C*) and indicated genes with known neutrophil functions, ranked first by % of overrepresented guides and then by median overrepresentation and mapped to a proteome list of human primary neutrophils. (**b–d**) Depicted are mean -/+ SEM of 3 independent experiments.

The online version of this article includes the following source data and figure supplement(s) for figure 1:

**Source data 1.** List of genes identified as hits in the CRISPR/Cas9 screen.
**Figure supplement 1.** Characterization of PLB-985 function and morphology.

We also analyzed two datasets comparing mRNA and protein expression of human primary bone marrow cells undergoing differentiation into neutrophils at various stages of maturation, namely myelocytes, metamyelocytes, band form neutrophils and segmented neutrophils (*Adams et al., 2012*; *Hoogendijk et al., 2019*) (see Materials and methods section for RNA-seq). Consistent with our PLB-985 data, we found a similar reduction of H1 mRNA as maturation progressed (*Figure 2—figure supplement 1c*). The protein levels of H1 subtypes in these datasets remained relatively stable during differentiation, despite a reduction in mRNA (*Figure 2—figure supplement 1c,d*). Hence, all H1 subtypes are expressed in differentiating primary neutrophils and although the mRNA levels decrease over time of differentiation, the protein levels seem to be relatively stable.

mRNA levels of each H1 subtypes were still detectable in differentiated PLB-985 and primary cells. Accordingly, human primary neutrophils isolated from blood of healthy donors expressed detectable amounts of the H1 genes. H1.2 and H1.4, the two hits in our genome wide screen, were the most abundantly expressed subtypes (*Figure 2b*). Interestingly, peripheral blood mononuclear cells (PBMCs) and monocytes showed a slightly different expression pattern of this gene family than neutrophils (*Figure 2—figure supplement 1e,f*).

To verify the results from our screen, we disrupted expression of H1.2 and H1.4 in PLB-985 by targeting these genes with CRISPR/Cas9. We used two clones deficient for H1.2 (from two sgRNAs) and two clones deficient for H1.4 (from one sgRNA) in our analysis. Cell death was reduced in both H1.2 and H1.4-deficient clones, confirming the results of the screen (*Figure 2c*). H1.2 and H1.4-deficient clones produced ROS less efficiently than wild type cells upon PMA stimulation (*Figure 2d*). Furthermore, cells deficient for either of the two H1 subtypes expressed less MPO (*Figure 2e*). Cells deficient for H1.2 or H1.4 could still phagocytose, demonstrating that not all effector functions were equally affected by the loss of H1 subtypes (*Figure 2—figure supplement 1g,h*). These findings suggest that H1.2 and H1.4 are required for PLB-985 to differentiate into mature, neutrophil-like cells.

## Opposing effects of H1 subtypes on neutrophil differentiation

To confirm that deficiency of H1.2 and H1.4 affected PLB-985 maturation, we showed that surface expression of the differentiation marker CD11b is decreased on both d3 and d7 of differentiation (*Figure 3a,b*, *Figure 3—figure supplement 1a,b*) in cells where these genes were deleted. We also disrupted expression of the other three somatic H1 subtypes. Surprisingly, loss of these subtypes resulted in the opposite phenotype. We observed markedly enhanced expression of CD11b already at d3, especially in clones deficient in H1.1 and H1.5 (*Figure 3a,b*, *Figure 3—figure supplement 1a, b*). To test whether these effects were specific to neutrophilic cells, we also disrupted expression of H1 subtypes in the monocytic cell line THP-1 and analyzed their ability to release IL-1β via the NLRP3 inflammasome. Batch populations of H1-deficient cells released similar amounts of IL-1β than control cells when activated with silica crystals (*Figure 3—figure supplement 1c*). We therefore conclude that loss of H1 subtypes does not generally impact myelopoiesis or innate immune functions.

Differentiation of neutrophils and PLB-985 is terminal - if not activated, these cells die by apoptosis. The number of wild type and control (scr.) PLB-985 cells increased more from d0 to d3 than from d3 to d7. This suggests that cells withdrew from proliferation (*Figure 3c*). Interestingly, H1.2 and H1.4-deficient clones grew unrestrictedly until d7, whereas cells deficient in H1.1, H1.3 and H1.5 decreased in comparison to controls (*Figure 3c,d*). Furthermore, control cells successively lost viability from d8 to d12 (*Figure 3e,f*). In line with their enhanced growth, H1.2 and H1.4-deficient cells

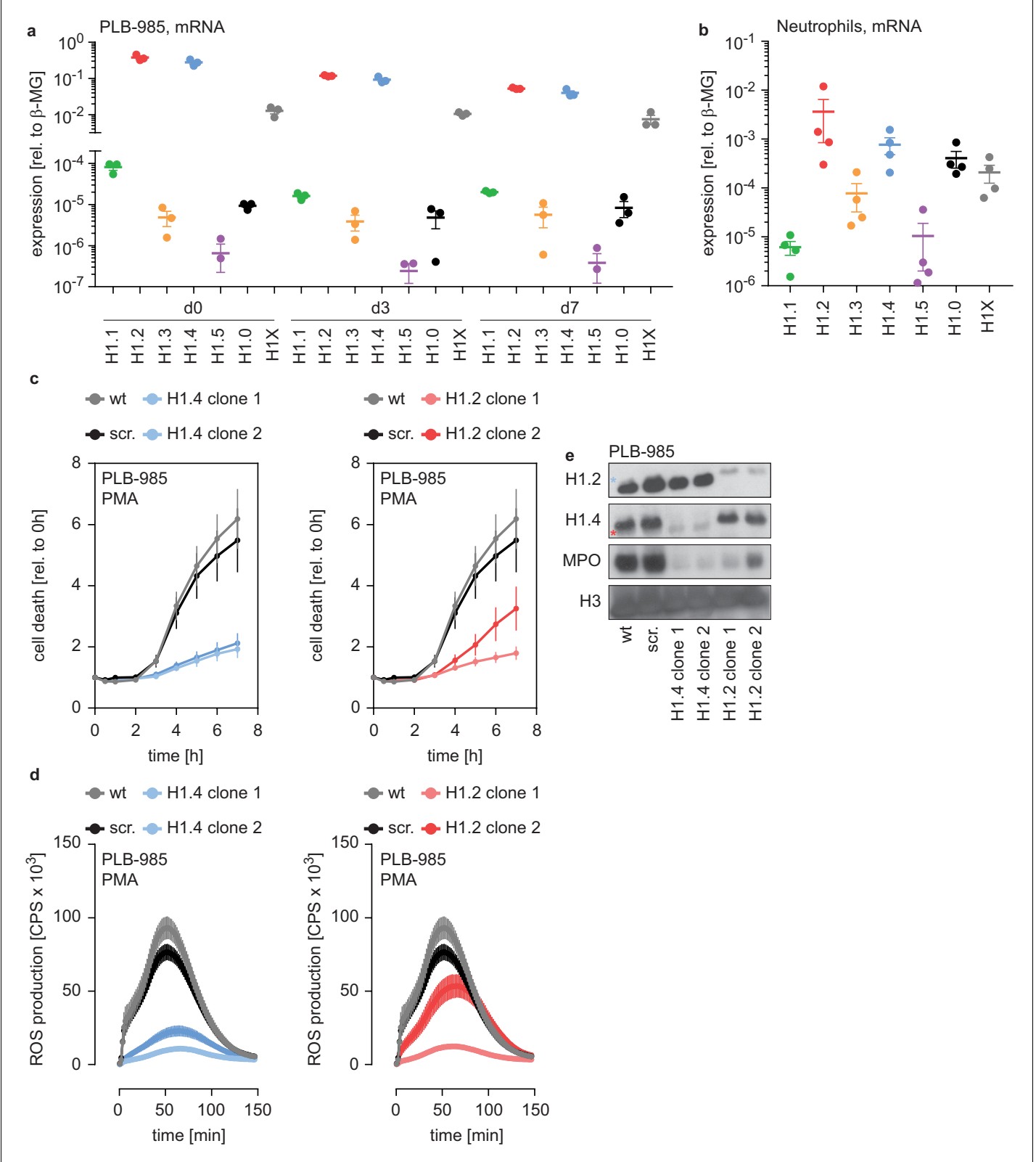

**Figure 2.** H1.2 and H1.4 are required for PLB-985 differentiation. (a) mRNA expression levels of indicated H1 subtypes (nomenclature refers to protein names) in PLB-985 cells at d0, d3 and d7 of differentiation, relative to the housekeeping gene β-microglobulin, depicted is the mean -/+ SEM of 3 independent experiments. (b) mRNA expression levels of indicated H1 subtypes in human primary neutrophils, relative to β-microglobulin. Data points are from four different donors, error bars are mean -/+ SEM. (c) PMA-induced cell death of wild type (wt) scrambled sgRNA (non-target, scr.) and two

*Figure 2 continued on next page*

*Figure 2 continued*

clones of either H1.2 or H1.4 over time, measured by SYTOX Green fluorescence. Depicted are mean -/+ SEM of 5 independent experiments of cells at d7 of differentiation. (d) Measurement of PMA-induced ROS production from wt, scr. and H1.2 or H1.4 deficient PLB-985, depicted are mean -/+ SEM of 5 independent experiments at d7 of differentiation. (c, d) wt and scr. values are the same in the left and right panels, respectively. (e) Western blot of lysates of PLB-985 at d3 of differentiation, showing efficient disruption of H1.2 and H1.4 (both antibodies recognize the other subtype, which is marked by a red asterisk for H1.2 and a blue asterisk for H1.4) as well as reduced MPO expression, the core histone H3 served as loading control.

The online version of this article includes the following figure supplement(s) for figure 2:

**Figure supplement 1.** H1 expression levels in human PBMCs, monocytes and PLB-985.

were more viable than controls, whereas H1.1, H1.3 and H1.5-deficient cells showed less viability than controls (*Figure 3e,f*). Of note, cells deficient in H1.1 had a stronger oxidative burst in response to PMA than control cells, consistent with enhanced maturation (*Figure 3—figure supplement 1d*). Clones deficient in H1.3 and H1.5, on the other hand, produced less ROS than control cells (*Figure 3—figure supplement 1e,f*). This could be a specific effect of H1.3 and H1.5, but it could also reflect reduced viability at the onset of the experiment. In summary, these experiments show an unexpected and opposing effect of H1 subtypes on PLB-985 cell differentiation and function.

## The impact of H1 deficiency manifests before onset of differentiation

To understand the extent of H1 subtype-specific effects on neutrophil differentiation we performed RNA-sequencing of the knockout lines. We sampled wild type cells, scr. cells and two clones per H1 subtype at d0, d3 and d7 of differentiation in four independent experiments (*Figure 4—source data 1*). To validate the applicability of our experiment, we compared the transcriptome of control PLB-985 cells (the combination of wild type and scr. samples) to human primary cells at various differentiation stages towards neutrophils (*Adams et al., 2012*) (also see Materials and methods section). Gene set enrichment analysis demonstrated that PLB-985 and primary cells concordantly regulate processes during differentiation (*Figure 4a*, *Figure 4—figure supplement 1a,b*. *Figure 4—figure supplement 1a* depicts the same analysis as *Figure 4a* with the gene set labels). Using principal component analysis (PCA), we saw that these cells follow a similar differentiation trajectory (*Figure 4—figure supplement 1c*). As expected, among the most robustly upregulated genes during differentiation of control cells we found several with known functions in neutrophils (*Figure 4b*).

We analyzed the H1-deficient clones by PCA and found that H1.2 and H1.4-deficient clones at d7 clustered more closely with control cells at d3, indicating delayed differentiation (*Figure 4c*). H1.1, H1.3 and H1.5-deficient cells at d3 clustered with control cells at d7, suggesting accelerated differentiation (*Figure 4c*).

Analysis of the enrichment of gene modules confirmed that, during differentiation, control cells upregulated genes related to the innate immune response and downregulated genes corresponding to cell cycle and division, in line with the context of terminal differentiation (*Figure 4—figure supplement 1d*). Interestingly, in H1.2 and H1.4-deficient cells, the modules were less regulated, again demonstrating prolonged proliferation and obstruction of differentiation (*Figure 4—figure supplement 1d*). We performed the same analysis for clones deficient in H1.1, H1.3 and H1.5 and confirmed accelerated differentiation (*Figure 4—figure supplement 1d*). We subsequently asked whether absence of H1 subtypes also affected the determination and lineage choice of PLB-985 before differentiation. To address this question, we compared differentially expressed genes of H1.2 and H1.4-deficient cells and control cells at d0. We found that both H1.2 and H1.4-deficient cells downregulated neutrophil genes, such as *MPO* and neutrophil proteases among others (*Figure 4d*), indicating that lineage fate was affected in undifferentiated cells.

Surprisingly, among the upregulated genes we found a significant number of genes associated with eosinophils and a GATA-1 signature (*Figure 4d,e*, see Materials and methods section for the source of datasets). We found this upregulation specifically in clones deficient in H1.2 and H1.4 in comparison to all other genotypes (*Figure 4e*). This suggests that loss of H1.2 and H1.4 affects lineage determination of PLB-985 cells even before the onset of differentiation by driving them towards a more eosinophil-like fate. Notably, we also saw dysregulation of C/EBP family transcription factors in cells deficient for different H1 subtypes (*Figure 4—figure supplement 2*). We found downregulation of *CEBPA* in cells deficient for H1.2 and H1.4 at d0, but upregulation of the same transcription

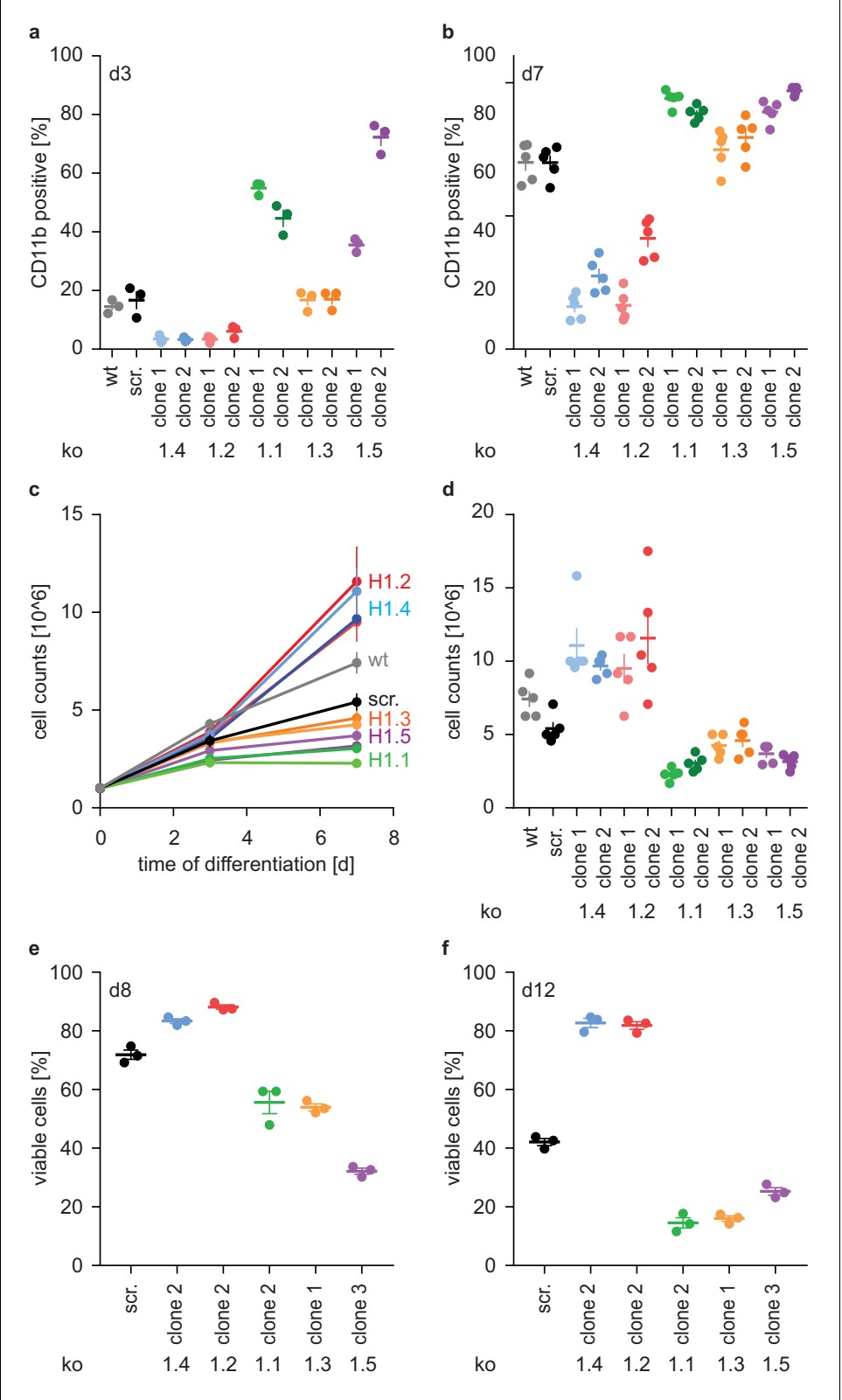

**Figure 3.** H1 effects on PLB-985 differentiation are subtype-specific. (**a, b**) Expression of the surface marker CD11b at d3 (**a**) or d7 (**b**) of differentiation. Depicted is the percentage of CD11b-positive cells out of all living singlets for wt, scrambled (scr.) and two knockout clones per H1 subtype. (**c, d**) Growth curves of PLB-985 during differentiation (**c**) and cell counts at d7 of differentiation (**d**) show restricted growth of H1.1, H1.3 and H1.5-

*Figure 3 continued on next page*

*Figure 3 continued*

deficient cells and enhanced growth of H1.2 and H1.4-deficient clones. (a–d) Depicted are five independent experiments, error bars are SEM. (e, f) Viability, as determined by DAPI negative cells, at d8 (e) and d12 (f) of differentiation of 3 independent experiments. Error bars are SEM.

The online version of this article includes the following figure supplement(s) for figure 3:

**Figure supplement 1.** Subtype-specific H1 impact on PLB-985 maturation and function.

factors in cells deficient for H1.1, H1.3 and H1.5 (*Figure 4—figure supplement 2*). This is in line with decreased and enhanced maturation, respectively. Other C/EBP family members also showed dysregulated expression patterns in H1-deficient clones (*Figure 4—figure supplement 2*). Our data indicate that H1 subtype depletion affects both cell fate determination and differentiation per se.

## Enhanced eosinophil differentiation of H1.2/H1.4-deficient hematopoietic stem cells

Loss of H1.2 and H1.4 appears to bias the lineage commitment of PLB-985 away from neutrophils via regulation of an eosinophil-like transcriptional program. We therefore analyzed the morphology of these cells by transmission electron microscopy. There was no obvious morphological difference between genotypes (*Figure 4—figure supplement 3a*). However, loss of H1.2 and H1.4 led to enhanced expression of the eosinophil marker Siglec-8 on the surface of a subset of PLB-985 (*Figure 4—figure supplement 3b*). H1.2- and H1.4-deficient PLB-985 also expressed higher levels of the eosinophil protein galectin-10/CLC, in line of what we had seen by RNA-seq (*Figure 4—figure supplement 3c,d*). We therefore conclude that even though PLB-985 do not become fully functional eosinophils in our differentiation system, H1.2 and H1.4 deficiency leads to enhanced expression of some eosinophil proteins. We also looked at H1 mRNA expression levels in human primary eosinophils and found that these cells expressed very high levels of H1.2 and H1.4 (*Figure 4—figure supplement 3e*). A proteomics dataset of cells in circulation confirmed high expression levels of H1 subtypes in eosinophils (*Figure 4—figure supplement 3f*; *Rieckmann et al., 2017*), suggesting that eosinophil maturation relies on an appropriate amount of H1 subtype expression during differentiation.

To test whether this eosinophil bias also exists in primary cells, we analyzed cells isolated from H1.2/H1.4 double-deficient mice (*Fan et al., 2003*). The profile of circulating lymphocytes and monocytes in the blood or bone marrow of adult animals was indistinguishable from wild type animals (*Figure 5—figure supplements 1* and *2*). Furthermore, wild type and H1.2/H1.4-deficient animals had the same proportion of neutrophils in circulation and in bone marrow (*Figure 5a,b*, *Figure 5—figure supplements 1* and *2*). We also induced a sterile peritonitis in these animals by injection of casein into the peritoneal cavity (*Zhang et al., 1997*). 24 hr upon injection the numbers of circulating leukocytes and bone marrow leukocytes were similar between genotypes (*Figure 5—figure supplement 3*).

Importantly, although the number of eosinophils in the bone marrow were similar between genotypes (*Figure 5—figure supplement 4a*), H1.2/H1.4-deficient animals had enhanced amounts of circulating eosinophils during homeostasis, suggesting that, in vivo, there is also bias towards eosinophils (*Figure 5c*). The number of circulating neutrophils returned to the same level as in wild-type animals upon casein injection (*Figure 5—figure supplement 4b,c*). In addition to a change in eosinophil numbers in circulation, H1.2/H1.4-deficient mice showed disturbances in neutrophil ageing markers (*Zhang et al., 2015*; *Adrover et al., 2019*). We found similar expression of CD62L in bone marrow neutrophils during homeostasis, but a slightly lower number of CD62L+ CXCR4-'young' neutrophils (*Zhang et al., 2015*; *Adrover et al., 2019*) in H1.2/H1.4-deficient mice (*Figure 5—figure supplement 5a–c,g*). Upon casein injection, we found more CD62L+, but also more CXCR4+ neutrophils in wildtype than in H1.2/H1.4-deficient animals (*Figure 5—figure supplement 5d–f,h*). These results suggest a slight disturbance of neutrophil ageing upon H1 deficiency. Interestingly, we also found altered expression of cytokines and chemokines. H1.2/H1.4-deficient bone marrow cells expressed enhanced levels of the chemokine CXCL1, reduced amounts of *IL6* mRNA, and we also found enhanced levels of IL-17 upon casein injection (*Figure 5—figure supplement 6a–d*). IL-17 affects granulopoiesis via G-CSF (*Stark et al., 2005*). Upon casein injection, the levels of

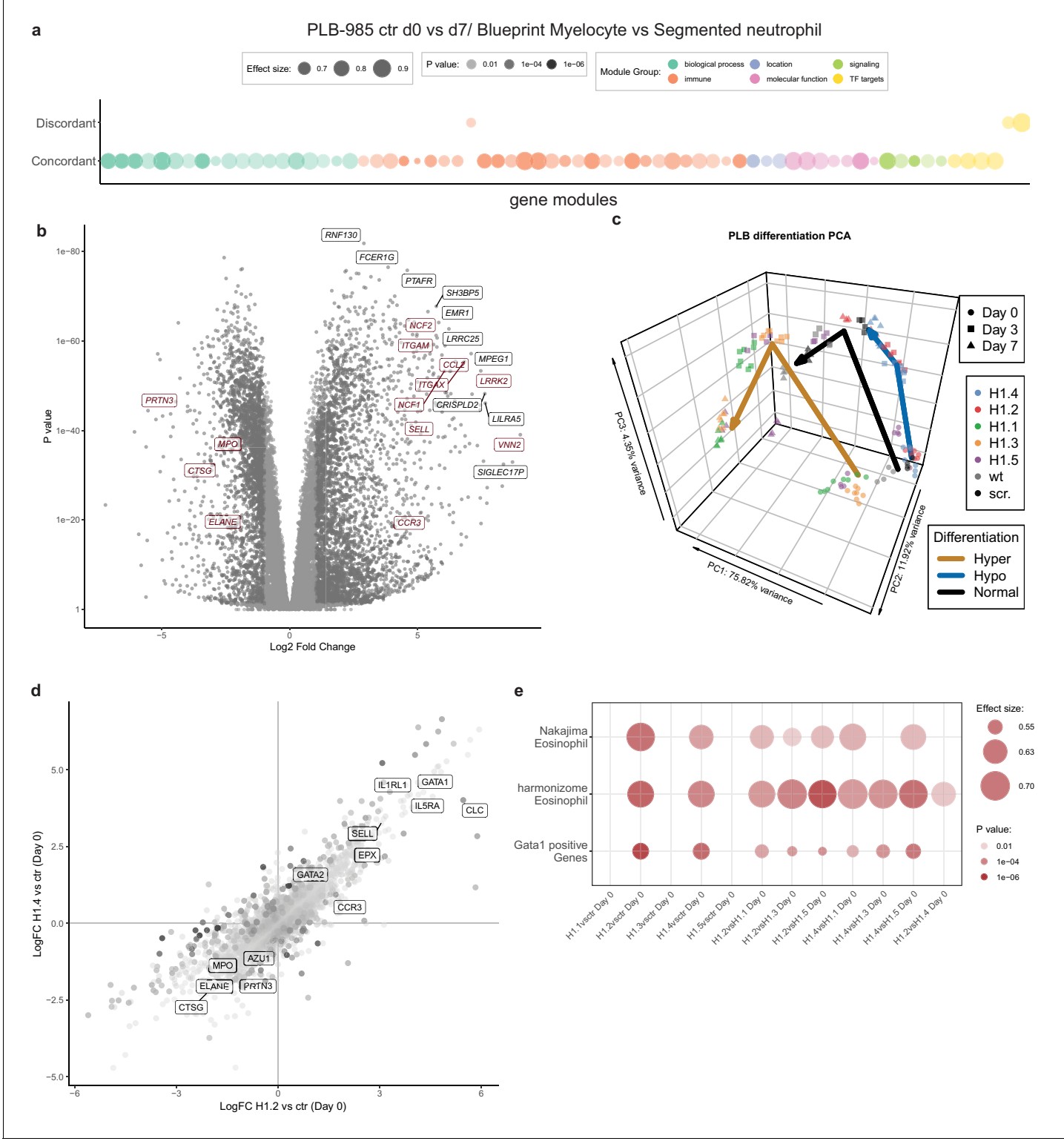

**Figure 4.** RNA-seq reveals H1 subtype-specific effects on differentiation and an eosinophilic signature of undifferentiated H1.2/H1.4 knockout lines. (**a**) Concordantly and discordantly regulated genes when comparing PLB-985 differentiation (wt and scr., d0 to d7) and human neutrophil differentiation (Blueprint, neutrophilic myelocyte to segmented neutrophil of bone marrow) using discordance scores and tmod. Most modules are concordantly regulated. (**b**) Volcano plot of all differentially expressed genes of control cells (wt and scr.) between d7and d0, darker gray depicts p<0.05 and logFC >2. A selection of strongly upregulated genes is labelled and neutrophil markers are highlighted in red. Expression of granule proteins, such as MPO, neutrophil elastase (*ELANE*), proteinase 3 (*PRTN3*) and cathepsin G (*CTSG*) decreases during differentiation. (**c**) Principal component analysis

*Figure 4 continued on next page*

Figure 4 continued

(PCA) of control and mutant PLB-985 during differentiation shows clustering according to experimental replicates and genotypes and separates samples according to differentiation state. Lines indicate the center-of-mass trajectory for 'normal' (wild type and scr. cells), 'hyper' (H1 clones with enhanced differentiation) and 'hypo' (H1 clones with reduced differentiation) samples. While lines deficient in H1.1, H1.3 and H1.5 already at d3 cluster closely to wt and scr. at d7, indicating faster and more efficient differentiation, H1.2 and H1.4-deficient clones at d7 still only cluster with wt and scr. at d3, indicating a stalled development.(d) Scatter dot plot depicting differentially expressed genes of H1.2 knockouts vs control cells at d0 of differentiation (x axis) and H1.4 knockouts vs control cells at d0 of differentiation (y axis), color (alpha) indicates p-value of differential expression between H1.2 and H1.4 at d0. Neutrophil genes such as neutrophil elastase (*ELANE*), proteinase 3 (*PRTN3*), cathepsin G (*CTSG*), azurocidin (*AZU1*) and *MPO* are strongly downregulated in both conditions when compared to wt and scr., whereas eosinophil genes such as galectin 10 (*CLC*), interleukin-5 receptor alpha (*IL5RA*), GATA-1 (*GATA1*), GATA-2 (*GATA2*) or eosinophil peroxidase (*EPX*) are upregulated. (e) Gene set enrichment analysis of indicated samples at d0 of differentiation shows enhanced expression of eosinophil and GATA-1 gene sets (see Materials and methods) in H1.2 and H1.4-deficient lines, but not in other knockout conditions.

The online version of this article includes the following source data and figure supplement(s) for figure 4:

**Source data 1.** RNA-seq expression tables.
**Figure supplement 1.** Transcriptional differences between H1-deficient clones during differentiation.
**Figure supplement 2.** Dysregulation of several transcription factors upon loss of H1 subtypes.
**Figure supplement 3.** Expression of eosinophil markers in PLB-985 upon loss of H1.2 or H1.4.

G-CSF were similar between wild type and H1.2/H1.4-deficient animals (*Figure 5—figure supplement 6e*). However, the response of bone marrow cells to G-CSF was altered upon loss of H1.2/ H1.4; these cells were more viable after overnight incubation in the presence of G-CSF than wild type cells (*Figure 5—figure supplement 7a*). Furthermore, bone marrow cells of H1.2/H1.4-deficient mice were more granular than wild type cells, which could reflect reduced spontaneous degranulation or differentially aged cells (*Adrover et al., 2020*; *Figure 5—figure supplement 7b*). We also looked at the granularity of neutrophils in circulation and in the bone marrow of animals during homeostatic and inflammatory conditions. H1.2/H1.4-deficient neutrophils were more granular in the bone marrow, both during homeostasis and upon casein injection, but not in circulation (*Figure 5— figure supplement 7c,d*). Taken together, we concluded that the altered cytokine and chemokine milieu in these mice might lead to compensatory effects on the maturation of neutrophils in vivo.

To analyze the intrinsic differentiation potential of H1.2/H1.4-deficient cells in the absence of confounding factors, we isolated lineage negative cells from murine bone marrow and differentiated them into several lineages (*Dahlin et al., 2018*). Importantly, H1.2/H1.4-deficient cells expressed elevated levels of surface markers for eosinophils and decreased levels of neutrophil surface markers, as compared to wild type cells. This recapitulates the phenotype we observed in PLB-985 (*Figure 5d–f*). These experiments show that H1.2 and H1.4 cooperate to determine the neutrophil-eosinophil cell fate decision in hematopoietic progenitor cells. Furthermore, our findings suggest that, in vivo, the H1.2 and H1.4 deficiency is compensated by a different turnover of neutrophils and by a dysregulated cytokine milieu to generate normal amounts of neutrophils during homeostasis. Importantly, the bias towards eosinophils also exists in vivo, demonstrating that loss of H1.2 and H1.4 affects the eosinophil lineage determination.

## Loss of H1.2 and H1.4 affect neutrophil cell fate determination via GATA-2

Two transcription factors required for eosinophil differentiation, GATA-1 and GATA-2, were upregulated on mRNA level in H1.2 and H1.4 knockout lines (*Figure 4d*). *GATA2* transcripts were higher than *GATA1* and transcript levels of both transcription factors at d0 inversely correlated with the potential of PLB-985 to adopt a neutrophil-like cell fate (*Figure 6—figure supplement 1a,b*). Interestingly, in mice *GATA1* mRNA expression was higher than *GATA2*, and we could also see enhanced expression of *GATA1* in H1.2/H1.4-deficient bone marrow cells as compared to wild type cells (*Figure 6—figure supplement 1c*). Expression of *GATA2* did not change in H1.2/H1.4-deficient murine cells, but we saw a slight downregulation of *CEBPA*, which is in line with the expression profiles of PLB-985 (*Figure 6—figure supplement 1d,e*).

We speculated that H1 subtypes affect neutrophil differentiation through either GATA-1 or GATA-2. To test this hypothesis, we disrupted expression of GATA-1 and GATA-2 by CRISPR/Cas9 in PLB-985 and analyzed batch populations of various mutations at d4 of differentiation. We chose

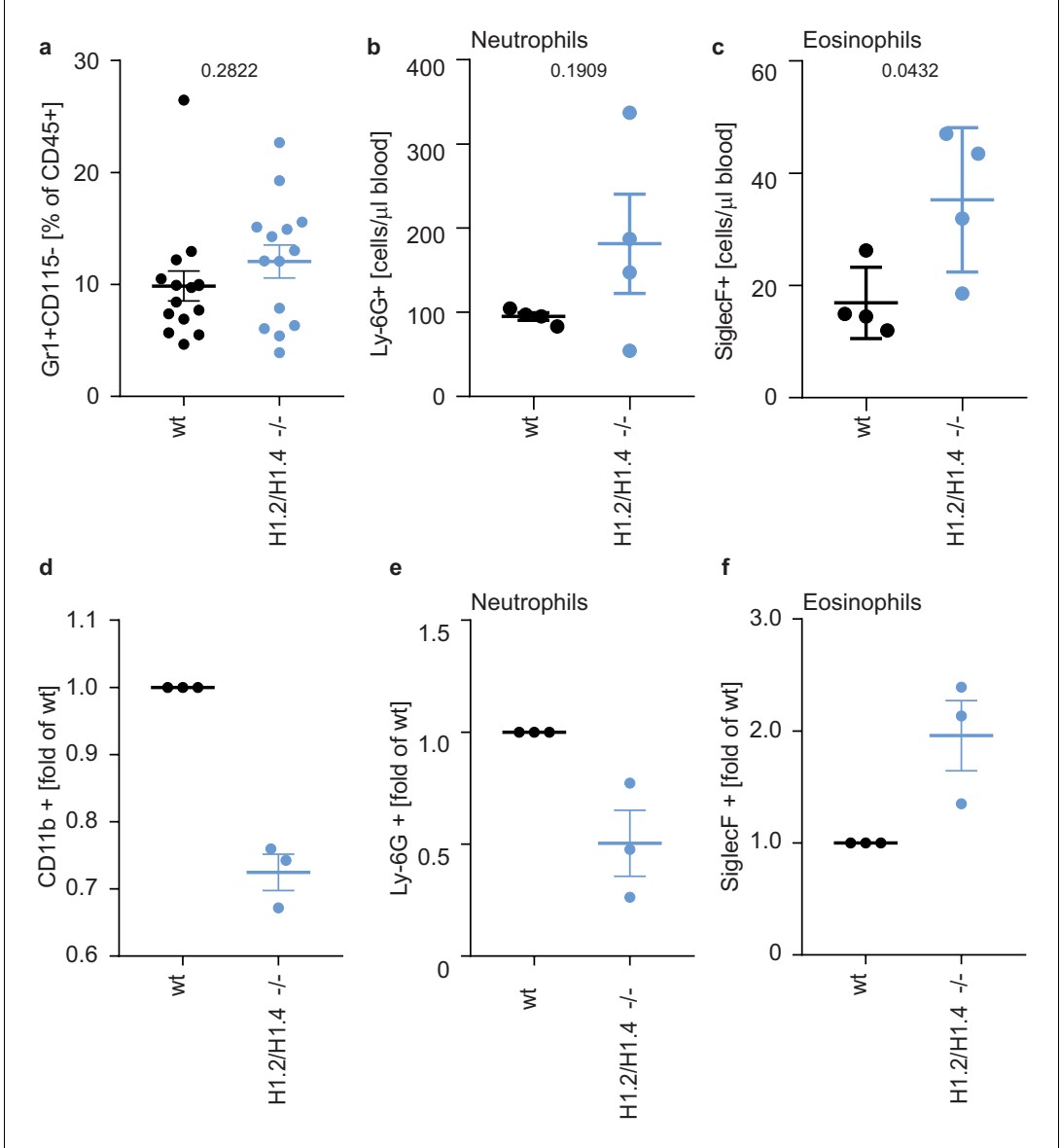

**Figure 5.** Deletion of H1.2 or H1.4 biases murine hematopoietic cells towards an eosinophil lineage. (a–c) Analysis of circulating neutrophils and eosinophils in wild type (wt) and H1.2/H1.4-double deficient mice (H1.2/H1.4 -/-). p Values, derived from unpaired two-tailed T tests, are indicated. (a) Flow cytometry analysis of neutrophils (Gr1 positive, CD115 negative) in whole blood of adult age-matched animals (wt n = 15, H1.2/H1.4 - / - n = 14), other immune cell types are shown in *Figure 5—figure supplement 1*. (b) Flow cytometry analysis of absolute amounts of neutrophils (Ly-6G positive cells) in whole blood (n = 4). (c) Flow cytometry analysis of amounts of eosinophils (SiglecF positive) in whole blood (n = 4). H1.2/H1.4 - / - animals show enhanced eosinophil numbers in circulation. d-f Lineage-negative hematopoietic stem cells were sorted from murine bone marrow and cultured for 6 days in the presence of various cytokines allowing differentiation into several immune cell lineages. Depicted are CD11b positive cells (d) and the fraction of neutrophils (e) or eosinophils (f) within CD11b-positive cells. Each dot represents an independent experiment consisting of two mice per genotype, the mean of the two wild type animals was set to one and the mean of the two H1.2/H1.4 - / - animals is depicted relative to wt. (a–f) Error bars are SEM.

The online version of this article includes the following figure supplement(s) for figure 5:

**Figure supplement 1.** Gating strategies and circulating leukocytes.

**Figure supplement 2.** Leukocytes in circulation and bone marrow in homeostasis.

**Figure supplement 3.** Leukocytes in circulation and bone marrow upon casein injection.

**Figure supplement 4.** Eosinophil counts in bone marrow upon casein injection and in circulation.

**Figure supplement 5.** Differentially regulated ageing markers in H1.2/H1.4-deficient mice.

**Figure supplement 6.** Differentially regulated cytokines in H1.2/H1.4-deficient mice.

**Figure supplement 7.** Enhanced survival and granularity of H1.2/H1.4-deficient bone marrow cells and neutrophils.

d4 because we assumed that cells still expressing GATA could outgrow GATA-deficient cells at later time points. Interestingly, GATA-2 deficiency enhanced maturation of PLB-985, as indicated by an increased ability to produce ROS in response to PMA (*Figure 6—figure supplement 2a*). In contrast, GATA-1 knockout lines behaved as scr. cells (*Figure 6—figure supplement 2a*).

To test whether the differentiation phenotype of H1.2 and H1.4-deficient lines relied on GATA-2, we performed a rescue experiment by generating double-deficient lines. We transduced scr. cells and H1.2 or H1.4-deficient cells with GATA-1 and GATA-2 sgRNAs or with a scr. sgRNA as a control and analyzed these populations at d4. Importantly, disruption of GATA-2 expression rescued ROS production (*Figure 6a–c*) and CD11b expression (*Figure 6d*) in H1.2 and H1.4-deficient PLB-985. Again, disruption of GATA-1 had no effect (*Figure 6a–d*). These experiments demonstrate that the differentiation phenotype resulting from H1.2 and H1.4 deficiency depends on upregulation of GATA-2 and, importantly, that loss of GATA-2 is sufficient to allow differentiation in the absence of these H1 subtypes.

In a second approach, we treated PLB-985 with inhibitors for GATA-1 and GATA-2, starting treatment one day before differentiation. Inhibition of GATA-2 enhanced CD11b expression of H1.2 or H1.4-deficient lines (*Figure 6e*). To confirm the effect of GATA-2 inhibition on differentiation, we measured mRNA expression of two genes, one that increased (aquaporin-9, *AQP9*) and one that decreased (*MPO*) during differentiation (*Figure 4—source data 1*). As expected, H1.2 and H1.4-deficient lines expressed less *AQP9* and more *MPO* than scr. cells at d7 of differentiation (*Figure 6—figure supplement 2b,c*). GATA-2 inhibition upregulated expression of *AQP9* and downregulated expression of *MPO* in H1.2 and H1.4 knockout lines at d7 of differentiation, indicating a rescue of maturation (*Figure 6—figure supplement 2b,c*). In line with our results with CRISPR/Cas9-mediated gene disruption, GATA-2 inhibition restored the capacity of H1.2 and H1.4-deficient clones to mount an oxidative burst (*Figure 6f–h*). As expected, treatment with the GATA-1 inhibitor did not affect maturation and function of PLB-985 of either genotype (*Figure 6e,i–k*). The combination of GATA-1 and GATA-2 inhibitors did not have an additive effect on PLB-985 maturation, confirming that GATA-2 is the relevant transcription factor for the phenotype observed in H1.2 and H1.4-deficient clones in the human cell system (*Figure 6—figure supplement 2d*). Importantly, the response to either GATA-2 deficiency or inhibition was stronger in H1-deficient PLB-985 than in scr. cells. This is in line with enhanced baseline expression of the transcription factor in H1.2 or H1.4-deficient clones. These data show that H1.2 and H1.4 affect fate determination and subsequent neutrophil differentiation and function at least partly through the regulation of GATA-2.

## Discussion

We identified an unexpected role of H1 subtypes in neutrophil development; loss of H1.2 and H1.4 inhibits differentiation while deficiency in H1.1, H1.3 and H1.5 enhances it. We propose that in neutrophil precursors, H1 subtypes control the expression or activity of GATA-2, which determines cell fate via transcription of neutrophil or eosinophil-specific genes. Enhanced expression of eosinophil genes – or reduced expression of neutrophil-specific genes – disrupts normal maturation, leading to prolonged proliferation and viability and reduced neutrophil function. Consistent with this, genetic ablation or inhibition of GATA-2, but not GATA-1, rescued the differentiation of H1.2 and H1.4-deficient cells. Interestingly, another study found that overexpression of GATA-2 in murine progenitor cells switched their lineage determination from macrophages to a megakaryocyte state (*Kitajima et al., 2002*). GATA-2 therefore acts as a key molecule for fate determination of immune cells.

It is noteworthy that we found evidence of enhanced *GATA1* mRNA expression in bone marrow cells of H1.2/H1.4-deficient mice, but not of *GATA2*. The fold change of *GATA1* in the absence of H1.2 and H1.4 was much higher in PLB-985 than the fold change of *GATA2*. Furthermore, in mice *GATA1* mRNA was higher expressed than *GATA2* mRNA. This could explain why we did not see an upregulation of *GATA2* in vivo. Another explanation could be that *GATA2* upregulation does not occur in lineage negative stem cells, but during the promyelocyte to neutrophil differentiation, while the regulation of *GATA1* is more robust. Enforced expression of both GATA-1 and GATA-2 can induce eosinophil differentiation of human CD34+ hematopoietic stem cells (*Hirasawa et al., 2002*). Interestingly, another study in murine cells showed that the eosinophil-promoting effect of GATA-2 depends strongly on the presence of C/EBP-α. Enforced expression of GATA-2 drove C/EBP-α

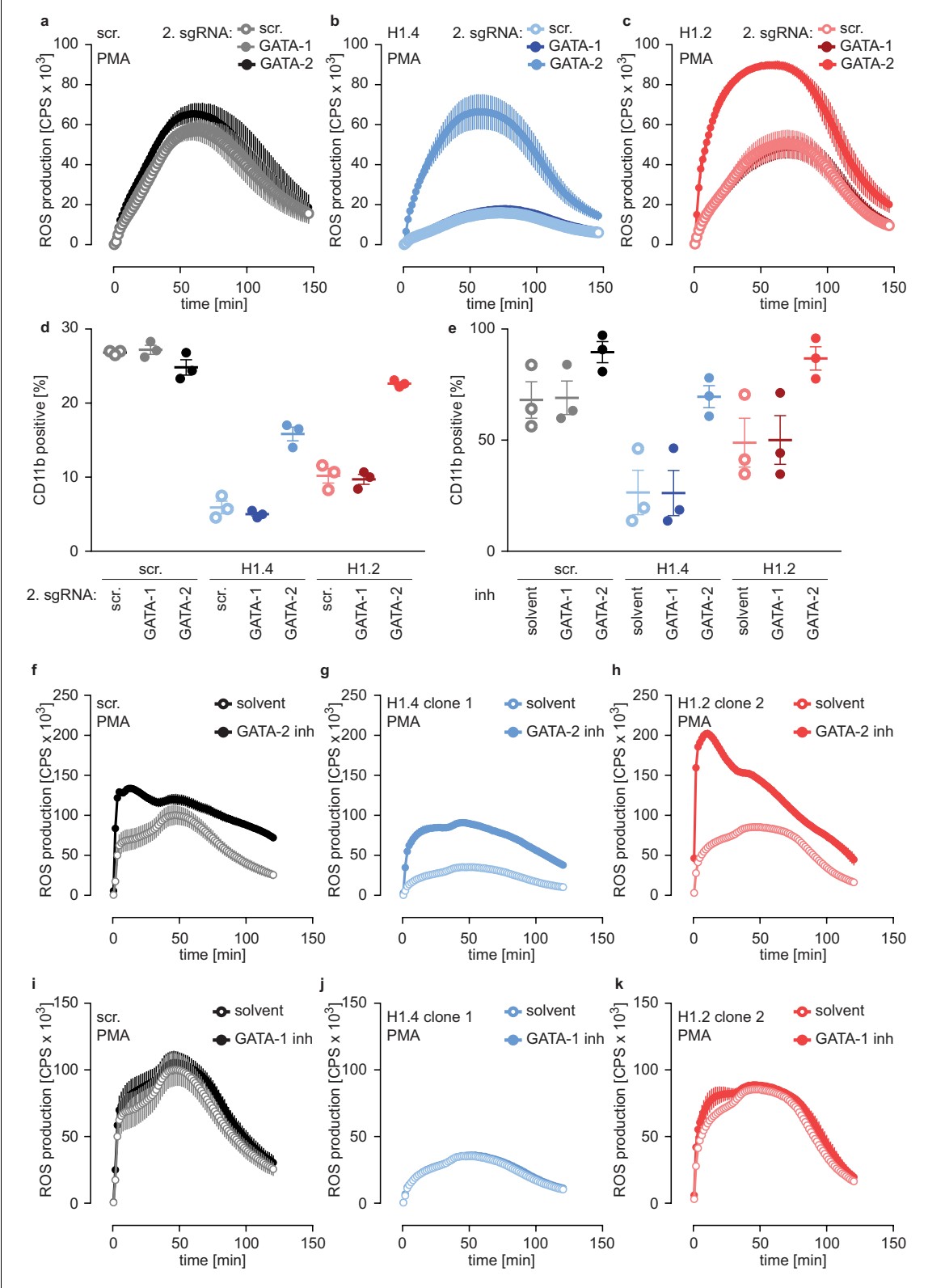

**Figure 6.** Deficiency of H1.2 or H1.4 is rescued by inhibition of GATA-2. (**a–c**) Deficiency in GATA-2 but not GATA-1 rescues H1.2 and H1.4 deficiency. Populations are batches of PLB-985. scr. (**a**), H1.4 (**b**) and H1.2 (**c**) knockout lines were transduced with an sgRNA against *GATA1* or *GATA2* or with a scr. sgRNA as control and analyzed for ROS production in response to PMA at d4 of differentiation. (**d**) Deficiency in GATA-2 rescues CD11b expression of H1.2 and H1.4-deficient PLB-985. CD11b expression of the indicated genotypes was measured at d4 of differentiation. (**e**) Treatment of PLB-985 with

*Figure 6 continued on next page*

*Figure 6 continued*

GATA-2 but not GATA-1 inhibitor rescues expression of CD11b, especially in lines deficient in H1.2 and H1.4. (**f–k**) ROS production of PLB-985 in response to PMA is affected upon GATA-2, but not GATA-1 inhibition during differentiation. (**f–k**) PLB-985 ROS production was measured at d7 of differentiation, indicated genotypes were treated with indicated inhibitors or solvent before the onset of differentiation and at d4. (**a–k**) Shown is the mean -/+ SEM of 3 independent experiments.

The online version of this article includes the following figure supplement(s) for figure 6:

**Figure supplement 1.** Expression of GATA-1, GATA-2 and C/EBP transcription factors in PLB-985 and murine bone marrow.

**Figure supplement 2.** H1.2 or H1.4 deficiency can be reverted by inhibition of GATA-2.

expressing precursor cells into the eosinophil lineage. However, in the absence of C/EBP-α, overexpression of GATA-2 induced mast cell and basophil development (*Iwasaki et al., 2006*). Therefore, not only the expression levels of hematopoietic transcription factors, but also the order of their expression is crucial for lineage determination. PLB-985 express C/EBP-α in the undifferentiated state, and enhanced expression of GATA-2 can therefore explain the upregulation of eosinophil markers we found in our experiments using this cell line.

Importantly, we observed a preference for eosinophil development in blood and in hematopoietic stem cells isolated from the bone marrow of H1.2/H1.4-deficient mice. This demonstrates that the bias towards the eosinophilic lineage also exists in vivo. However, adult H1.2/H1.4-deficient animals had normal numbers of circulating neutrophils, indicating that neutrophil differentiation is compensated. Interestingly the turnover rate or the longevity of neutrophils seemed to be affected in these mice, since we found reduced amounts of young neutrophils in H1.2/H1.4-deficient animals. We also found enhanced granularity and longevity of neutrophils isolated from H1.2/H1.4-deficient animals. These two findings, reduced levels of CD62L (a marker of young neutrophils) and enhanced granularity, are not consistent with the paradigm of young neutrophils being more granular (*Adrover et al., 2020*). Our data therefore point to a dysregulation of neutrophil behavior and/or turnover. Interestingly, however, the protein levels of H1.2 and H1.4 changed in human neutrophils during the time of day, suggesting that linker histone levels might affect neutrophil ageing in vivo (*Adrover et al., 2020*).

Neutropenia in peripheral tissues triggers de novo production of neutrophils in the bone marrow (*Stark et al., 2005*), demonstrating that granulopoiesis in vivo is regulated by feedback loops. This suggests that initial neutrophil deficiency in adult H1.2/H1.4-deficient mice is compensated by a different cytokine milieu. In line with this, we found altered expression of various cytokines and chemokines, thus suggesting a dysregulated turnover or production of neutrophils in H1.2/H1.4-deficient mice. It is noteworthy that we did not detect differences in the major cytokine driving neutrophil differentiation, G-CSF. Casein-induced peritonitis upregulated G-CSF in both wt and H1.2/H1.4-deficient animals. The serum levels of G-CSF during homeostasis were very low and probably under the detection limit of our ELISA, and, if there are minor differences in G-CSF levels, we were not able to measure them. We did, however, detect differences in the neutrophil-recruiting chemokine CXCL1 and in the pro-inflammatory cytokine IL-17, which could both also affect neutrophil turnover and behavior.

Interestingly, in human breast cancer cells, combined loss of H1.2 and H1.4 leads to an upregulation of an interferon signature (*Izquierdo-Bouldstridge et al., 2017*). Although we did not observe an obvious interferon signature in H1.2 or H1.4-deficient PLB-985, it is possible that in H1.2/H1.4-deficient mice some cell types contribute to immune cell differentiation via interferon-stimulated genes.

H1 can act in redundant and subtype-specific ways (*Hergeth and Schneider, 2015*). Mice lacking one or two H1 subtypes are viable and fertile (*Fan et al., 2003*). Analogously, PLB-985 deficient in any of the five H1 subtypes that we disrupted were viable and did not show growth defects. This is in contrast to another study, where shRNA-mediated knock down of H1.2 and H1.4 reduced cell growth and viability in various non-immune cell lines (*Sancho et al., 2008*). We found that loss of H1.2 and H1.4, in the context of neutrophil differentiation, enhanced viability, demonstrating cell type specific effects of H1 subtypes.

H1 has long been considered a general repressor of transcription (*Schlissel and Brown, 1984*). Accordingly, a study in human cells found that genomic regions associated with active transcription

were devoid of H1 subtypes (*Izzo et al., 2013*). However, murine embryonic stem cells that are triple-deficient in H1.2, H1.3 and H1.4 show a surprisingly low number of only 38 differentially expressed genes when compared to wild type cells (*Fan et al., 2005*). Our data show that loss of H1 subtypes in more differentiated cells affects the expression of many genes with clear functional consequences. We postulate that only a fraction of differentially expressed genes in PLB-985 is directly regulated by H1 subtypes. The enhanced expression of lineage-determining transcription factors, such as GATA-2, leads to further changes in the transcriptome and affects cell differentiation.

An important open question is: how do H1 subtypes control expression of GATA-2? It is possible that H1 subtypes directly occupy the promoter or enhancer elements of *GATA2* and silence its expression. H1 subtype-specific binding to genomic regions has, for example, been shown for human H1.5 (*Li et al., 2012*). Interestingly, H1.5 targeted over 20-fold more regions in differentiated human lung fibroblasts than in embryonic stem cells and H1.5 binding led to chromatin compaction and gene silencing. It is therefore tempting to speculate that similar mechanisms are at play during neutrophil development. In addition to a direct repressive effect of H1.5, its binding to chromatin was also associated with an enrichment of demethylation of histone H3 at lysine 9 (H3K9me2) (*Li et al., 2012*). H3K9me2 is a posttranslational modification of a core histone that is associated with a repressive chromatin state. H1 subtypes are therefore likely to regulate transcription via epigenetic marks on core histones. Furthermore, H1 subtypes themselves are posttranslationally modified in a subtype-specific manner (*Hergeth and Schneider, 2015*). Although less is known about H1 modifications than about core histones, it is likely that such modifications can determine subtype-specific protein-protein interactions, chromatin binding properties or other features that subsequently cause transcriptional changes.

It is puzzling that vertebrates express so many different H1 subtypes. Although they are able to compensate each other in 'extreme' situations, such as the loss of one or even two subtypes in a whole organism (*Fan et al., 2003*), they may also have more subtle and specific functions. This study contributes to our understanding of how different H1 subtypes initiate distinct transcriptional programs. Importantly, these programs appear to be evolutionarily conserved, since we found similar responses in human and murine cells.

Our observations in the human cell line PLB-985 and murine stem cells ex vivo correlate with an increase in the number of circulating eosinophils in H1.2/H1.4-deficient mice in vivo. We also detected a disturbed cytokine milieu and a dysregulation of neutrophil ageing markers, granularity and longevity in H1.2/H1.4-deficient animals, suggesting altered neutrophil turnover. This would be in line with altered expression levels of H1.4 and H1.2 during human neutrophil ageing in circulation (*Adrover et al., 2020*).

Notably, the number of neutrophils in the bone marrow and in circulation in H1.2/H1.4-deficient mice was normal. Moreover, eosinophil numbers in bone marrow were the same between genotypes. Taking all these observations and limitations together, we propose that H1 subtype regulation in vivo is subject to compensatory mechanisms that eventually adjust the levels of neutrophils. Indeed, it is expected that deletion of 2 out of 5 H1 subtypes, that probably affect the epigenetic regulation of development, are subject to systemic adjustments. Further in vivo work, ideally also aimed at investigating possible species differences between human and mouse, will likely elucidate the relevance of H1 deficiency on granulopoiesis and neutrophil turnover in vivo during homeostasis and inflammation.

In summary, we identified novel and opposing functions for specific H1 subtypes in granulopoiesis. H1 subtypes affect the lineage determination between neutrophils and eosinophils. We speculate that inappropriate expression of H1 subtypes in the hematopoietic compartment shapes pathologies with neutrophil or eosinophil involvement, such as asthma. Our findings suggest that linker histones represent another layer of complexity to the fascinating process of immune cell differentiation.

## Materials and methods

### Cell lines

PLB-985 were kindly provided by Prof. Mary Dinauer, Washington University School of Medicine (*Zhen et al., 1993*). THP-1 were from DSMZ-German Collection of Microorganisms and cell cultures.

The cell lines were authenticated by STR authentication (eurofins Genomics); PLB-985 and THP-1 wildtype strains were mycoplasma negative (eurofins Genomics). The generated knockout lines were not routinely tested for mycoplasma.

## Antibodies and staining reagents

H1.2 antibody (PA5-32009, Invitrogen), H1.4 antibody (41328S, Cell Signaling), GAPDH antibody (Cell Signaling, 2118), MPO antibody (A0398, Dako), Histone H3 antibody (ab1791, Abcam), galectin-10 antibody (Abcam, 231964), SYTOX Green nucleic acid stain (Thermo Fisher Scientific) and 4′,6-diamidino-2-phenylindole (DAPI; Sigma) were used. All flow cytometry antibodies were from BD Biosciences: human CD11b (557321), human Siglec-8 (347105), mouse CD11b (562605), mouse Ly-6G (551461), mouse SiglecF (565527), mouse CD45 (561487), mouse CD3 (561798), mouse CD4 (552775), mouse CD8a (561093), mouse CD115 (565249), mouse GR1 (561103).

## Chemicals and cytokines

Luminol (11050, AAT-Bioquest), horseradish peroxidase (31941, Serva), PMA (P8139, Sigma), A23187 (Santa Cruz Biotechnology Inc), GATA-1 inhibitor (anagrelide, SML0846, Sigma), GATA-2 inhibitor (K-7174, HY-12743A, MedChemExpress) were used. Recombinant murine cytokines (IL-3, IL-5, IL-9, GM-CSF, SCF, G-CSF) were from Peprotech.

## Plasmids

All CRISPR/Cas9 gene disruptions were done using lentiCRISPRv2 (*Shalem et al., 2014*; *Sanjana et al., 2014*). LentiCRISPRv2 was a gift from Feng Zhang (Addgene plasmid #52961; http://n2t.net/addgene:52961; RRID:Addgene_52961). Lentiviral particles were produced using psPAX and pMD2.G packaging plasmids in HEK293T cells.

## sgRNA sequences and primers

All sgRNA sequences and primer sequences are listed in *Supplementary file 1*.

## Donor consent

Human primary neutrophils, eosinophils, PBMCs and monocytes were isolated from blood samples of healthy volunteers according to the declaration of Helsinki. All donors provided written informed consent and all blood samples were collected from blood donated anonymously and with approval from the ethics committee–Charité –Universitätsmedizin Berlin (EA1/0104/06).

## Mice

Breeding of mice and isolation of blood and bone marrow were approved by the Berlin state authority Landesamt für Gesundheit und Soziales (license numbers for breeding: ZH161, ZH122, license numbers for isolation of cells: T0049/18, license number for injection of casein: H0085/18). All mice were bred at the Max Planck Insitute for Infection Biology under specific pathogen-free conditions. Animals were maintained on a 12 hr light/12 hr dark cycle and fed ad libitum. H1.2/H1.4-deficient animals were provided by Prof. Arthur Skoultchi (*Fan et al., 2003*). Casein injection was performed as described (*Zhang et al., 1997*). Briefly, mice were injected intraperitoneally with 1 ml of 7% casein in the evening and again 12 hr later. 3 hr after the second injection, animals were sacrificed and blood as well as bone marrow was isolated for analysis.

## Primary cell isolation

Neutrophils were purified by centrifugation of whole blood over Histopaque-1119 (Sigma), followed by centrifugation over a discontinuous Percoll gradient (*Zhang et al., 1997*). PBMCs were collected from the upper phase after Histopaque-1119 centrifugation, washed and either used for RNA extraction or for isolation of monocytes by positive selection (CD14) with magnetic beads (130-050-201, Miltenyi Biotec). Eosinophils were isolated from full blood using an isolation kit according to the manufacturer's instructions (130-092-010, Miltenyi Biotec).

## PLB-985 propagation and differentiation

PLB-985 were cultured in RPMI (RPMI 1640 medium, gibco) supplemented with 10% FCS, L-glutamine and antibiotics (25030–024 and 15140–122, gibco), cells were passaged every 3–4 days. For differentiation, cells were counted at d0 and $0.4 \times 10^6$ cells/ml were seeded in differentiation medium (RPMI supplemented with 2.5% FCS, 0.5% dimethylformamide (DMF, D4551, Sigma), L-glutamine and antibiotics), usually in six well plates in a total volume of 3 ml. At d4 of differentiation, 2 ml of differentiation medium were added per six well. At d7, cells were washed and passed over an equal volume of Histopaque-1077 (Sigma) (centrifugation for 20 min at 800 x g, brakes off) to remove debris and subsequently seeded at the required density for experiments in assay medium (RPMI (without phenol red) containing 1% FCS, 0.5% DMF and L-glutamine).

## Transmission electron microscopy

Cell suspensions, fixed in 2.5% glutaraldehyde, were sedimented and embedded in low melting agarose (2% in PBS). Drops of agarose containing cells were then postfixed in 0,5% osmiumtetroxide, contrasted with tannic acid and 2% uranyl acetate, dehydrated in a graded ethanol series and embedded in epoxy resin. After polymerization, sections were cut at 60 nm and contrasted with lead citrate. Specimens were analyzed in a Leo 906E transmission electron microscope at 100KV (Zeiss, Oberkochen, DE) using a side mounted digital camera (Morada; SIS-Olympus Münster DE).

## Analysis of ROS production from cells

Cells were seeded at $10^5$/well of a white 96 well plate (Nunc) in triplicates in a final volume of 100 µl in assay medium. HRP (1.2 U/ml) and luminol (50 µM) were added 10 min after seeding and 10 min later cells were stimulated with 100 nM PMA. Luminescence signal was measured using a VICTOR XLight Multimode Plate Reader (Perkin Elmer).

## Analysis of cell death

Cells were seeded at $10^5$/well of a 96 well plate in duplicates or triplicates in a final volume of 100 µl. The cell impermeable DNA dye SYTOX Green (50 nM) was added and 10 min later cells were stimulated with 100 nM PMA or with 5 µM A23187. SYTOX Green fluorescence signal corresponding to NET formation and cell death was measured over time using excitation/emission of 485/518 nm. Alternatively, microscopy pictures were acquired using brightfield and GFP channels (for SYTOX Green) in order to visualize the morphology of dead cells.

## Analysis of phagocytosis

PLB-985 at d7 of differentiation were seeded at $5 \times 10^5$ in a final volume of 20 µl and stimulated with a multiplicity of infection (MOI) 5 of fluorescently labelled (GFP-expressing or Alexa Fluor 488 conjugated) *E. coli* (Thermo Fisher Scientific). After incubation for 30 min at 37°C on a shaker (300 rpm), the cells were washed twice with PBS and acquired by flow cytometry for the uptake of bacteria.

## Analysis of surface marker expression

PLB-985 and murine ex vivo differentiated bone marrow cells were stained at various time points of differentiation with indicated surface marker antibodies for 20 min at 4°C in PBS with 2% FCS, washed and analyzed by flow cytometry using a MACS Quant Analyzer (Miltenyi Biotec). DAPI was added just before acquisition to exclude dead cells.

Murine blood was stained the same way as described above, but after staining, samples were incubated with eBioscience 1 Step Fix/Lyse Solution (invitrogen, 00-5333-54) for 1 hr to lyse erythrocytes before acquisition.

## CRISPR/Cas9 screen

PLB-985 were transduced with a lentiviral library containing CRISPR/Cas9 sgRNAs (*Shalem et al., 2014*) at an estimated MOI of <1. Cells were selected with puromycin (Sigma, 2.5 µg/ml), differentiation was performed as described for wt PLB-985. At d7 of differentiation, cells were stimulated with 100 nM PMA for 16 hr and surviving cells were sorted by FACS. DNA extraction was performed by DNeasy Blood and Tissue Kit (Qiagen) and PCR amplification was performed as described

(*Shalem et al., 2014*). Reads were mapped to the library and abundance of sgRNAs was calculated per condition. Genes with greater than 50% of their sgRNAs identified at values of at least 2-fold overrepresentation in the PMA-treated group versus the control group were defined as hits.

## Generation of CRISPR/Cas9 knockout lines

PLB-985 were transduced with lentiviral particles containing the sgRNA of interest cloned into lenti-CRISPRv2, cells were selected with puromycin (2.5 µg/ml) and, if single clones were analyzed, seeded at a density of 0.8 cells/well in 96 wells for limited dilution cloning. Clones were sequenced using the OutKnocker protocol and software (*Schmid-Burgk et al., 2014*; *Supplementary file 1*). Clones with out of frame indels leading to efficient gene disruption were used for subsequent experiments.

For GATA-1 and GATA-2 disruption in scr., H1.2 and H1.4-deficient cells, sgRNAs targeting GATA were cloned into a modified lentiCRISPRv2, in which the Cas9 sequence was exchanged for a blasticidin resistance cassette. After transduction, cells were selected with blasticidin (50 µg/ml) and used in batch populations for experiments.

## RNA extraction and qRT-PCR

Cell samples were pelleted, resuspended and lysed in TRIzol Reagent (life technologies). RNA was isolated by chloroform extraction and isopropanol precipitation. Similar amounts of total RNA were used for RT-PCR (High Capacity cDNA Reverse Transcription Kit, 4368814, Thermo Fisher Scientific). qRT-PCR was performed using SYBR Green (4385612, Thermo Fisher Scientific) on a QuantStudio 3 (Thermo Fisher Scientific).

## RNA-sequencing

Total RNA of PLB-985 at various time points was isolated using RNeasy kit (Qiagen). Library preparation and illumina sequencing were performed by the Max Planck-Genome-centre Cologne, Germany (https://mpgc.mpipz.mpg.de). The data were mapped to hg38.87 using STAR 2.5.2b (*Dobin et al., 2013*) and differential expression analysis was performed using edgeR (*McCarthy et al., 2012*). Gene level read counts from primary human cells at various steps of neutrophil differentiation were retrieved from Blueprint (*Adams et al., 2012*) (http://www.blueprint-epigenome.eu, Datasets EGAD00001002446, EGAX00001244028, EGAX00001244022 and EGAD00001002366) and analyzed the same way. Gene module enrichment was performed using tmod package (*Weiner and Domaszewska, 2016*) and concordance and discordance was analyzed by calculating discordance scores (*Domaszewska et al., 2017*). Eosinophil gene sets were retrieved from Harmonizome (*Rouillard et al., 2016*) and Broad Institute (http://software.broadinstitute.org/gsea, Nakajima set; *Nakajima et al., 2001*), GATA-1 gene set was generated by associating GATA-1 ChIP seq peaks (GEO accession GSM1003608) to genes using ChIPseeker (*Yu et al., 2015*).

## Ex vivo differentiation of murine bone marrow cells into neutrophils and eosinophils

Murine bone marrow was extracted from the femurs and tibiae of adult age-matched mice. Cells were flushed, erythrocytes were lysed by osmotic lysis, cells were washed and lineage-negative precursor cells were sorted by negative selection using magnetic beads (130-110-470, Miltenyi Biotec). Lineage-negative cells were subsequently propagated analogous to the protocol described (*Dahlin et al., 2018*). Briefly, cells were seeded in IMDM supplemented with 20% FCS, antibiotics, L-glutamine and 0.1 µM 2-mercaptoethanol (Sigma), as well as with the recombinant cytokines IL-3 (20 ng/ml), IL-5 (50 ng/ml), IL-9 (50 ng/ml), GM-CSF (10 ng/ml) and SCF (20 ng/ml) for 6 days.

## Statistical analysis

The descriptive statistics of all figure panels are mentioned in the respective figure legends. Populations of circulating leukocytes in murine blood were compared by a two-tailed, unpaired t test and p values are indicated above the populations. RNA-seq data were analyzed as described in the respective methods section.

## Acknowledgements

We thank Dr. Volker Brinkmann, Ulrike Abu Abed and Christian Goosmann for their help with microscopy experiments and we thank the Max Planck Genome-centre Cologne for performing the sequencing of the PLB-985 samples in this study. We also thank Dr. Alf Herzig, Dr. Borko Amulic and Anna Zychlinsky Scharff for reading and commenting on the manuscript. GS was funded by an Early Postdoc.Mobility and an Advanced Postdoc.Mobility fellowship from the Swiss National Science Foundation. Arthur I Skoultchi was funded by an NIH H1 grant (GM116143). The project was funded by the Max Planck Society.

## Additional information

### Funding

| Funder | Grant reference number | Author |
| --- | --- | --- |
| Schweizerischer Nationalfonds zur Förderung der Wissenschaftlichen Forschung | P300P3_158518 | Gabriel Sollberger |
| Schweizerischer Nationalfonds zur Förderung der Wissenschaftlichen Forschung | P2EZP3_148748 | Gabriel Sollberger |
| Max-Planck-Gesellschaft | | Gabriel Sollberger Robert Streeck Brian Edward Caffrey Arturo Zychlinsky |
| National Institute of General Medical Sciences | GM116143 | Arthur I Skoultchi |

The funders had no role in study design, data collection and interpretation, or the decision to submit the work for publication.

### Author contributions

Gabriel Sollberger, Conceptualization, Data curation, Formal analysis, Funding acquisition, Validation, Investigation, Visualization, Writing - original draft, Project administration, Writing - review and editing; Robert Streeck, Conceptualization, Software, Formal analysis, Validation, Visualization, Methodology; Falko Apel, Data curation, Formal analysis, Investigation, Methodology, Writing - review and editing; Brian Edward Caffrey, Data curation, Formal analysis, Validation, Visualization; Arthur I Skoultchi, Resources, Writing - review and editing; Arturo Zychlinsky, Conceptualization, Data curation, Supervision, Funding acquisition, Project administration

### Author ORCIDs

Gabriel Sollberger https://orcid.org/0000-0003-3647-9714

### Ethics

Human subjects: Human primary neutrophils, PBMCs and monocytes were isolated from blood samples of healthy volunteers according to the declaration of Helsinki. All donors provided written informed consent and all blood samples were collected with approval from the local ethics committee.

Animal experimentation: Breeding of mice and isolation of blood and bone marrow were approved by the Berlin state authority Landesamt für Gesundheit und Soziales. All mice were bred at the Max Planck Insitute for Infection Biology under specific pathogen-free conditions. Animals were maintained on a 12-hour light/12-hour dark cycle and fed ad libitum.

### Decision letter and Author response

Decision letter https://doi.org/10.7554/eLife.52563.sa1
Author response https://doi.org/10.7554/eLife.52563.sa2

# Additional files

## Supplementary files
• Supplementary file 1. List of sgRNA sequences, primer sequences and antibodies. Individual sheets contain sequences of sgRNAs, sequencing primers, qRT-PCR primers and antibody catalog and lot numbers.

• Supplementary file 2. Key Resources table. Table of reagents, cell lines, genetically modified organisms, others.

• Transparent reporting form

## Data availability
RNA sequencing data have been deposited in ArrayExpress - accession no. E-MTAB-8459. All data generated or analysed during this study are included in the manuscript and supplementary files. Source data files are provided for Figure 1 and Figure 4. A supplementary file with all used qPCR primers, sgRNA sequences, antibodies and other reagents is provided.

The following dataset was generated:

| Author(s) | Year | Dataset title | Dataset URL | Database and Identifier |
|---|---|---|---|---|
| Sollberger G, Streeck R, Apel F, Caffrey BE, Skoultchi AI, Zychlinsky A | 2020 | RNA-seq of human neutrophil-like cell line PLB-985 at various stages of differentiation with CRISPR knock-outs of H1 linker histones | https://www.ebi.ac.uk/arrayexpress/experiments/E-MTAB-8459/ | ArrayExpress, E-MTAB-8459 |

The following previously published datasets were used:

| Author(s) | Year | Dataset title | Dataset URL | Database and Identifier |
|---|---|---|---|---|
| Blueprint | 2016 | BP_August_2016_RNA-Seq_band_form_neutrophil_on_GRCh38 - samples | http://dcc.blueprint-epigenome.eu/#/datasets/EGAD00001002446 | Blueprint DCC, EGAD00001002446 |
| Blueprint | 2016 | neutrophilic myelocyte from bone marrow of donor: BM230614, BM220513 | http://dcc.blueprint-epigenome.eu/#/datasets/EGAX00001244028 | Blueprint DCC, EGAX0000124402 8 |
| Blueprint | 2016 | segmented neutrophil of bone marrow from bone marrow of donor: BM230614, BM220513 | http://dcc.blueprint-epigenome.eu/#/datasets/EGAX00001244022 | Blueprint DCC, EGAX0000124402 2 |
| Blueprint | 2016 | BP_August_2016_RNA-Seq_neutrophilic_metamyelocyte_on_GRCh38 - samples | http://dcc.blueprint-epigenome.eu/#/datasets/EGAD00001002366 | Blueprint DCC, EGAD0000100 2366 |
| Nakajima T, Matsumoto K, Suto H, Tanaka K, Ebisawa M, Tomita H, Yuki K, Katsunuma T, Akasawa A, Hashida R, Sugita Y, Ogawa H, Ra C, Saito H | 2001 | NAKAJIMA_EOSINOPHIL | https://www.gsea-msigdb.org/gsea/msigdb/cards/NAKAJIMA_EOSINOPHIL | Molecular Signatures Database, NAKAJIMA_EOSINOPHIL |
| ENCODE DCC | 2012 | Stanford_ChipSeq_K562_GATA1_(SC-266)_IgG-mus | https://www.ncbi.nlm.nih.gov/geo/query/acc.cgi?acc=GSM1003608 | NCBI Gene Expression Omnibus, GSM1003608 |
| van den Biggelaar M | 2019 | Dynamic transcriptome-proteome correlation networks reveal human myeloid differentiation and neutrophil-specific programming | http://proteomecentral.proteomexchange.org/cgi/GetDataset?ID=PXD013785 | ProteomeXchange, PXD013785 |

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
