## [Decision Letter]

**Acceptance summary:**

The editors and reviewers want to highlight the importance of the CRISPR screen established here to identify components required for normal neutrophilic differentiation of PLB-985 cells. Your description of the novel role for histone H1 subtypes in regulating neutrophil-lineage specification and differentiation versus eosinophilic/basophilic lineages is highly novel.

**Decision letter after peer review:**

Thank you for submitting your article "Linker Histone H1 subtypes specifically regulate neutrophil differentiation" for consideration by *eLife*. Your article has been reviewed by Satyajit Rath as the Senior Editor, a Reviewing Editor, and two reviewers. The reviewers have opted to remain anonymous.

The reviewers have discussed the reviews with one another and the Reviewing Editor has drafted this decision to help you prepare a revised submission.

Summary:

Sollberger et al., used a CRISPR screen to identify components required for normal neutrophilic differentiation of PLB-985 cells. They found that deletion of linker histones H1.2 and/or H1.4 reduced the differentiation of the cells towards a neutrophilic phenotype. Further experiments revealed that the absence of H1.2/H1.4 led to skewing of the cells towards GATA2-mediated differentiation to eosinophilic/basophilic lineages. Analysis of H1.2/H1.4 double knockout hematopoietic stem cells revealed that their in vitro differentiation was skewed towards eosinophils rather than neutrophils.

Essential revisions:

Both reviewers recognized interest and novelty in this study, describing a novel role for histone H1 subtypes in regulating neutrophil-lineage specification and differentiation in an in vitro differentiation setting. However, a major concern raised both at the first review and post-review by both reviewers as well as a major concern for the editor lies in the translation into a in vivo setting in mouse models or human cells. Indeed, no reduction in the circulating neutrophil counts in H1.2/H1.4 double knockout mice was observed, reducing the biological significance of the work beyond its technical novelty and could even suggest artifactual findings of the in vitro setting developed by the authors. The authors should perform an in depth analysis of bone marrow and blood levels (not percentage) of neutrophils in steady state and inflammation (for example repopulation after bone marrow transplant or marrow proliferating cell ablation or in a model of emergency granulopoiesis).

As previously mentioned, neutrophil lineage specification is determined by transcription factors such as CEBP-α or – ε. How H1.2 and H1.4 function? Are CEBP-α or – ε levels also altered in cells deficient for H1.2 and H1.4 or is it only dependent of GATA-2? Do cells deficient for H1.2 and H1.4 exhibit phenotypic and functional characteristics similar to eosinophils? Are these proteins expressed at low levels in primary eosinophils? Is GATA-2 upregulated in the bone marrow of mice deficient for H1.2 and H1.4?

[Editors' note: further revisions were suggested prior to acceptance, as described below.]

Thank you for sending your article entitled "Linker histone H1.2 and H1.4 affect the neutrophil lineage determination" for peer review at *eLife*. Your article is being evaluated by Satyajit Rath as the Senior Editor, a Reviewing Editor, and two reviewers.

The essential point of discussion was the need to revise the analysis of age-young neutrophils based on CD62L expression (second question) to potentially support your hypothesis, plus additional major textual revisions. About the major textual revisions, we all recognized the value of the CRISPR screen in cell from the neutrophil-lineage which is a substantial achievement and the attempt to have made effort to add more in vivo relevance. However, the reviewers made the point that those experiments do not fit as logically into the manuscript as the original results and that the in vivo results are still confusing. Hence, we would also request to clarify the presentation and interpretation of the new data, with a more thorough and objective discussion on the fact that the major observation was made on a neutrophil cell line, that the in vivo confirmation needs further work and more discussion for the discrepancies found in this study (increase in circulating eosinophils but not with changes in neutrophil numbers, no differences in G-CSF level, lower amount of young neutrophils in the bone marrow…).

Please see below the comments that were raised in more details.

Reviewer #1:

In this revision, Sollberger and colleagues provide new data to address the in vivo importance of histone H1 during neutrophil-lineage specification and differentiation by analyzing in depth granulopoiesis in H1.2/H1.4-deficient mice during homeostasis and inflammation. Interestingly, in homeostasis -not during casein-induced peritonitis- absolute numbers of eosinophils are increased in blood of mutant mice, which is agreement with the in vitro data. However, as observed in the initial manuscript, neutrophil absolute numbers are not affected (or even increased) both under steady state and during inflammation. The authors claim that altered cytokine profile involved in granulopoiesis (IL-17a) and neutrophil mobilization (CXCL1) and changes in neutrophil maturation (reduced number of young neutrophils and increased granularity) may explain these results observed in the H1.2/H1.4-deficient mice.

For the reviewer these statements are not clear:

- First, no changes in G-CSF are found which could be associated to increase granulopoiesis that could compensate neutrophil numbers.

- Second, a lack of neutrophils due to altered granulopoiesis in H1.2/H1.4-deficient mice should generate de novo production of neutrophils, hence observing increased levels of young neutrophils in the bone marrow, not decreasing.

Maybe I misunderstood but it is not clear to me how Figure 5—figure supplement 2B and 2C can be possible. If absolute numbers of neutrophils (2A) are not altered and the percentage of CD62l+ is not affected either, how in 2C are decreased. This concern also applies to graphs 2D-F. How CD62L percentage was calculated? The authors should show the plots of these results (not only the gating strategy).

- According to Casanova-Acebes et al., (2013), aged neutrophils exhibit low granularity and size. If lower numbers of young neutrophils are found and hence more aged are present, a decreased in SSC and FSC should be seen. Not an increase.

- How are the numbers of eosinophils in the bone marrow? This result is of key importance and it is not shown.

- How are the levels of eosinophils in blood and bone marrow after casein treatment?

I agree with the authors that many confounding factors can influence the results in the mutant mice but still such results are key for the message of the paper. Based on the observed results, no strong effect of H1 subtypes is observed on neutrophil differentiation in homeostasis and during inflammation. Maybe such difference could have been seen during marrow regeneration (i.e. after 5-fluorouracil treatment).

Reviewer #2:

This is an interesting work on a CRISPR screen on neutrophil-like cell lines identifying a role for linker histones on neutrophil differentiation.

The authors have appropriately addressed the reviewers' and the editor's questions.

---

## [Author Response]

Essential revisions:Both reviewers recognized interest and novelty in this study, describing a novel role for histone H1 subtypes in regulating neutrophil-lineage specification and differentiation in an in vitro differentiation setting. However, a major concern raised both at the first review and post-review by both reviewers as well as a major concern for the editor lies in the translation into a in vivo setting in mouse models or human cells. Indeed, no reduction in the circulating neutrophil counts in H1.2/H1.4 double knockout mice was observed, reducing the biological significance of the work beyond its technical novelty and could even suggest artifactual findings of the in vitro setting developed by the authors. The authors should perform an in depth analysis of bone marrow and blood levels (not percentage) of neutrophils in steady state and inflammation (for example repopulation after bone marrow transplant or marrow proliferating cell ablation or in a model of emergency granulopoiesis).

We thank the reviewers and the editor for their suggestions and we agree that a more detailed analysis of granulopoiesis is important. We analysed wild type and H1.2/H1.4-deficient animals during steady state and after Casein injection into the peritoneal cavity, which induces a peritonitis and leads to the influx of neutrophils (Amulic et al., 2017). In the new version of the manuscript, as suggested, we include counts of leukocytes rather than percentage (new Figure 5B,C, new Figure 5—figure supplement 2, Figure 5—figure supplement 3, Figure 5—figure supplement 4). We showed, as detailed below, two main changes in mice deficient in H1.2/H1.4; the dysregulation of cytokine/chemokine and a change in neutrophil ageing markers and granularity.

Importantly, we also found an increased number of circulating eosinophils at steady state in H1.2/H1.4-deficient animals (new Figure 5C). This is in line with our findings in human cells and shows that the loss of H1.2 and H1.4 affects the neutrophil/eosinophil lineage determination in vivo.

Interestingly, we found that both during steady state and upon casein injection, H1.2/H1/4-deficient mice produced altered amounts of cytokines and chemokines compared to wild type animals (new Figure 5—figure supplement 4H-L). Most prominently, H1.2/H1.4-deficient bone marrow cells transcribed less IL-6 and more CXCL1, the latter is an important neutrophil chemokine (new Figure 5 —figure supplement 4H,I). We also found this upregulation of CXCL1 on the protein level in serum (new Figure 5—figure supplement 4J) and we further found enhanced production of IL-17 in the serum of casein-injected H1.2/H1.4-deficient mice compared to wild type animals (new Figure 5—figure supplement 4K). IL-17 is part of a feed forward loop leading to granulopoiesis (Stark et al., 2005) via the production of G-CSF, though we did not detect altered G-CSF levels in H1.2/H1.4-deficient animals (new Figure 5—figure supplement 4L). Taken together, our results indicate that granulopoiesis in H1-deficient animals is altered due to changes in cytokines and chemokines.

We further analysed the neutrophil population in H1.2/H1.4-deficient mice using a second panel of surface marker antibodies by flow cytometry (new Figure 5—figure supplement 1B). The combined use of CD62L and CXCR4 allows the analysis of young vs aged neutrophils (Adrover et al., 2019; Zhang et al., 2015) and we found reduced numbers of young neutrophils in the bone marrow of H1.2/H1.4-deficient animals (new Figure 5—figure supplement 4A-C) suggesting compensatory effects regulating neutrophil turnover in these mice.

Lastly, when we incubated bone marrow cells overnight in the presence or absence of G-CSF, we found that G-CSF induced increased survival in H1.2/H1.4-deficient cells compared to wild type cells (new Figure 5—figure supplement 5A). The H1-deficient bone marrow cells also were more granular than wild type cells after overnight incubation (new Figure 5—figure supplement 5B), indicating that the spontaneous degranulation is affected. Indeed, we also found H1.2/H1.4-deficient bone marrow neutrophils to be more granular than wild type cells (new Figure 5—figure supplement 5C,D). This would again be in line with a different ageing phenotype, since a recent publication showed spontaneous degranulation of neutrophils upon ageing (Adrover et al., 2020).

Taken together, we provide evidence of cytokine/chemokine dysregulation in H1-deficient mice, as well as a change in neutrophil ageing markers and granularity. We therefore think that our analysis of lineage negative stem cells isolated from bone marrow is an appropriate way of analysing the intrinsic differentiation capacity of H1-deficient cells in the absence of confounding factors.

As previously mentioned, neutrophil lineage specification is determined by transcription factors such as CEBP-α or – ε. How H1.2 and H1.4 function? Are CEBP-α or – ε levels also altered in cells deficient for H1.2 and H1.4 or is it only dependent of GATA-2? Do cells deficient for H1.2 and H1.4 exhibit phenotypic and functional characteristics similar to eosinophils? Are these proteins expressed at low levels in primary eosinophils? Is GATA-2 upregulated in the bone marrow of mice deficient for H1.2 and H1.4?

The reviewers point that other transcription factors might also affect neutrophil vs eosinophil differentiation in our system, is, of course, interesting. Indeed, we found in our RNA-seq experiment that C/EBP-α levels are downregulated in PLB-985 deficient for H1.2 or H1.4 at the onset of differentiation (new Figure 4—figure supplement 2). In cells deficient for H1.1, H1.3 and H1.5 expression of C/EBP-α was enhanced at the onset of differentiation, which is in line with the fact that these cells showed accelerated maturation (new Figure 4—figure supplement 2). We subsequently analysed the expression of other C/EBP family transcription factors. C/EBP-α expression was enhanced in H1.2- and H1.4-deficient cells at later time points of differentiation, but not at early time points. We also found enhanced expression of other C/EBP family members, most prominently an upregulation of C/EBP-α and C/EBP-α in clones deficient for H1.1, H1.3 and H1.5 (new Figure 4—figure supplement 2). We therefore conclude that, besides GATA-1 and GATA-2, other transcription factors are differentially regulated.

To assess the expression levels of H1 subtypes in human primary eosinophils, we isolated RNA from these cells and performed qRT-PCR experiments. Surprisingly, expression levels of H1.2 and H1.4 were very high in primary eosinophils, much higher than in any other cell type we studied (new Figure 4—figure supplement 3E). This indicates that H1.2 and H1.4 levels are indeed crucial for adequate eosinophil differentiation. We assume that too high as well as too low expression of these subtypes affect the lineage determination of neutrophils and eosinophils.

When we analysed GATA expression in the bone marrow of H1.2/H1.4-deficient mice, we found that these animals showed enhanced expression of GATA-1, in line with what we found in PLB-985 (new Figure 6—figure supplement 1C). GATA-2 expression was neither changed in RNA isolated from total bone marrow nor in lineage negative stem cell RNA isolations (new Figure 6—figure supplement 1d), which could be due to the altered cytokine milieu in vivo. We also noted that in mice expression of GATA-1 was higher than of GATA-2 (new Figure 6—figure supplement 1C,D), opposite to what we had found in human cells. It could therefore be that the loss of H1.2 and H1.4 in mice more readily affects the higher expressed GATA-1.

Taken together, we provide evidence that, besides GATA transcription factors, C/EBP family transcription factors are also dysregulated in H1-deficient cells in human (new Figure 4—figure supplement 2) and mouse (new Figure 6—figure supplement 1E) and we show an effect of H1.2 and H1.4 deficiency on GATA-1 in vivo.

Regarding the question whether PLB-985 become eosinophils in the absence of H1.2 or H1.4, we believe that in PLB-985 cells, the loss of H1.2 or H1.4 affects eosinophil markers primarily on a transcriptional level, since H1.2- and H1.4-deficient clones remained viable and the cells divided at late stages of differentiation (Figure 3C-F). We did, however, analyse the morphology and function of PLB-985 to assess whether they became functional eosinophils. We did not observe specific eosinophil morphology of H1.2- or H1.4-deficient PLB-985 cells by transmission electron microscopy (TEM) (new Figure 4—figure supplement 3A). However, when we looked at eosinophil markers on the protein level, we did observe an increase in the expression of Siglec-8 on the surface of (new Figure 4—figure supplement 3B) H1.2- or H1.4-deficient cells and we also found enhanced expression of the eosinophil protein galectin-10/CLC (new Figure 4—figure supplement 3C,D). We therefore conclude that deficiency of H1.2 or H1.4 also induces upregulation of eosinophil proteins, without affecting the overall morphology of PLB-985.

[Editors' note: further revisions were suggested prior to acceptance, as described below.]The essential point of discussion was the need to revise the analysis of age-young neutrophils based on CD62L expression (second question) to potentially support your hypothesis, plus additional major textual revisions. About the major textual revisions, we all recognized the value of the CRISPR screen in cell from the neutrophil-lineage which is a substantial achievement and the attempt to have made effort to add more in vivo relevance. However, the reviewers made the point that those experiments do not fit as logically into the manuscript as the original results and that the in vivo results are still confusing. Hence, we would also request to clarify the presentation and interpretation of the new data, with a more thorough and objective discussion on the fact that the major observation was made on a neutrophil cell line, that the in vivo confirmation needs further work and more discussion for the discrepancies found in this study (increase in circulating eosinophils but not with changes in neutrophil numbers, no differences in G-CSF level, lower amount of young neutrophils in the bone marrow…).

We apologize for the apparent lack of clarity. We present our in vivo data in a more understandable way. Specifically, we rearranged Figure 5—figure supplement 4, Figure 5—figure supplement 5 to more figure panels with a better panel structure (new Figure 5—figure supplement 4, Figure 5—figure supplement 5, Figure 5—figure supplement 6, Figure 5—figure supplement 7). Moreover, we discuss the limitations of our study (Discussion section) and clearly state that more work will have to be done regarding the in vivo relevance of our findings.

Please see below the comments that were raised in more details.Reviewer #1:In this revision, Sollberger and colleagues provide new data to address the in vivo importance of histone H1 during neutrophil-lineage specification and differentiation by analyzing in depth granulopoiesis in H1.2/H1.4-deficient mice during homeostasis and inflammation. Interestingly, in homeostasis -not during casein-induced peritonitis- absolute numbers of eosinophils are increased in blood of mutant mice, which is agreement with the in vitro data. However, as observed in the initial manuscript, neutrophil absolute numbers are not affected (or even increased) both under steady state and during inflammation. The authors claim that altered cytokine profile involved in granulopoiesis (IL-17a) and neutrophil mobilization (CXCL1) and changes in neutrophil maturation (reduced number of young neutrophils and increased granularity) may explain these results observed in the H1.2/H1.4-deficient mice.For the reviewer these statements are not clear:- First, no changes in G-CSF are found which could be associated to increase granulopoiesis that could compensate neutrophil numbers.

We thank the reviewer for acknowledging our new in vivo work. As the reviewer correctly points out, we did not detect changes in G-CSF (new Figure 5—figure supplement 6E) in homeostatic conditions or after injection of casein.

The levels of G-CSF in homeostatic conditions were very low. This means that, in homeostatic conditions, we are probably at the detection limit of the ELISA and cannot detect subtle differences if they exist. Upon an inflammatory stimulus, Casein, we detected G-CSF in circulation, but there were no differences between wild type and H1.2/H1.4 null mice.

It is relevant to point out that, while G-CSF is a major inducer of granulopoieses, this process is likely affected by other cytokines. Indeed, we observed changes in CXCL1 and IL17. We discuss the levels of G-CSF in H1.2/H1.4-deficient mice more critically in our revised manuscript on Discussion section.

- Second, a lack of neutrophils due to altered granulopoiesis in H1.2/H1.4-deficient mice should generate de novo production of neutrophils, hence observing increased levels of young neutrophils in the bone marrow, not decreasing.

We agree with the reviewer. However, we did see decreased levels of young (CD62L positive, CXCR4 negative) neutrophils (Figure 5—figure supplement 4C, new Figure 5—figure supplement 5). We discuss these findings (subsection “Enhanced eosinophil differentiation of H1.2/H1.4-deficient hematopoietic stem cells”, Discussion section) and acknowledge that they do not fit with the expected paradigm.

Maybe I misunderstood but it is not clear to me how Figure 5—figure supplement 2B and 2C can be possible. If absolute numbers of neutrophils (2A) are not altered and the percentage of CD62l+ is not affected either, how in 2C are decreased. This concern also applies to graphs 2D-F. How CD62L percentage was calculated? The authors should show the plots of these results (not only the gating strategy).

We think that the reviewer refers to Figure 5—figure supplement 4B and 4C. We apologize for the unclear plotting, 4B shows CD62L positive cells, whereas 4c shows CD62L positive and CXCR4 negative cells. We strongly believe that an inclusion of all individual flow cytometry plots to a supplementary figure (with 48 plots for Figure 5—figure supplement 4A-F alone) will make it impossible to understand the figure panels. Therefore, we rearranged our plots and we now include histograms of CD62L and CXCR4 expression where we overlay all animals as shown in our new Figure 5—figure supplement 5B,C,E,F.

- According to Casanova-Acebes et al., (2013), aged neutrophils exhibit low granularity and size. If lower numbers of young neutrophils are found and hence more aged are present, a decreased in SSC and FSC should be seen. Not an increase.

The reviewer is right to point out that ageing of neutrophils leads to reduces granularity, as also shown in 2020 by the same group (Adrover et al.,). Interestingly, the 2020 study by Adrover et al., also found changes of H1.2 and H1.4 on protein level as neutrophils aged. We cite this recent study and rephrase our wording in the manuscript (Discussion section). The fact that we find more neutrophils with ageing markers, but an enhanced survival and granularity after overnight incubation (new Figure 5—figure supplement 7) does not point towards an increase in ageing, but to a dysregulation of neutrophil behaviour, which we discuss in subsection “Enhanced eosinophil differentiation of H1.2/H1.4-deficient hematopoietic stem cells” and the Discussion section.

- How are the numbers of eosinophils in the bone marrow? This result is of key importance and it is not shown.- How are the levels of eosinophils in blood and bone marrow after casein treatment?

We include eosinophil numbers in blood and bone marrow, during homeostasis and upon casein injection in our revised manuscript (new Figure 5—figure supplement 4).

I agree with the authors that many confounding factors can influence the results in the mutant mice but still such results are key for the message of the paper. Based on the observed results, no strong effect of H1 subtypes is observed on neutrophil differentiation in homeostasis and during inflammation. Maybe such difference could have been seen during marrow regeneration (i.e. after 5-fluorouracil treatment).

We discuss our in vivo results more critically in the new version of the manuscript (Discussion section). The phenotypes we observed, as expected, are not strong. This is probably due to two reasons: first, we used animals deficient for, most likely, epigenetic modifiers rather than key transcription factors and, second, granulopoiesis is an important process that is regulated by feedback loops and via different cytokines. Another, third, reason could be that the compensatory effects are different or stronger upon deletion of two H1 subtypes in mouse versus the PLB-985, which are deficient for only one H1 subtype.